# HIV-1 targets L-selectin for adhesion and induces its shedding for viral release

Joseph Kononchik [1], Joanna Ireland[1], Zhongcheng Zou[1], Jason Segura[1], Genevieve Holzapfel[1], Ashley Chastain [1], Ruipeng Wang[1], Matthew Spencer[1], Biao He[1], Nicole Stutzman[1], Daiji Kano[1], James Arthos[2], Elizabeth Fischer[3], Tae-Wook Chun[2], Susan Moir [2] & Peter Sun[1]

CD4 and chemokine receptors mediate HIV-1 attachment and entry. They are, however, insufficient to explain the preferential viral infection of central memory T cells. Here, we identify L-selectin (CD62L) as a viral adhesion receptor on CD4$^+$ T cells. The binding of viral envelope glycans to L-selectin facilitates HIV entry and infection, and L-selectin expression on central memory CD4$^+$ T cells supports their preferential infection by HIV. Upon infection, the virus downregulates L-selectin expression through shedding, resulting in an apparent loss of central memory CD4$^+$ T cells. Infected effector memory CD4$^+$ T cells, however, remain competent in cytokine production. Surprisingly, inhibition of L-selectin shedding markedly reduces HIV-1 infection and suppresses viral release, suggesting that L-selectin shedding is required for HIV-1 release. These findings highlight a critical role for cell surface sheddase in HIV-1 pathogenesis and reveal new antiretroviral strategies based on small molecular inhibitors targeted at metalloproteinases for viral release.

---

[1] Laboratory of Immunogenetics, National Institute of Allergy and Infectious Diseases, National Institutes of Health, 12441 Parklawn Drive, Rockville, MD 20852, USA. [2] Laboratory of Immunoregulation, National Institute of Allergy and Infectious Diseases, National Institutes of Health, 9000 Rockville Pike, Bethesda, MD 20892, USA. [3] Research Technology Branch, National Institute of Allergy and Infectious Diseases, National Institutes of Health, 903 South 4th Street, Hamilton, MT 59840, USA. These authors contributed equally: Joseph Kononchik, Joanna Ireland. Correspondence and requests for materials should be addressed to P.S. (email: psun@nih.gov)

Human immunodeficiency virus type 1 (HIV-1) infection remains a major public health issue. In the absence of an effective vaccine, viral infection can only be managed by highly active antiretroviral treatment (HAART)[1]. To date, much of our understanding of HIV-1 entry is based on viral envelope proteins (gp120 and gp41) interacting with CD4 and chemokine receptors[2,3]. However, the role of gp120-associated glycans in HIV infection and pathogenesis is less clear. While glycosylations on gp120 shields the virus from humoral immune recognition[4,5], the viral glycans are often recognized by host lectin receptors, such as mannose receptor (MR), DEC-205, and DC-SIGN on macrophage and dendritic cells leading to viral capture and antigen presentation[6–9]. Some of the lectin receptors, such as Siglec receptors on macrophages, are also used by the virus to facilitate its adhesion and infection[10,11]. As these lectin receptors are not expressed on CD4+ T cells, it is not clear if HIV envelope glycans contribute to the viral infection of T cells despite earlier studies showing mutations in gp120 glycans resulted in replication deficient viruses[4].

While HIV-1 infects all CD4+ T cells, it exhibits a preference for central memory CD4+ T cells ($T_{CM}$)[12–14], and may target them for viral reservoirs[15–18]. L-selectin, also known as CD62L, is a marker for central memory T cells ($T_{CM}$). It facilitates lymphocyte rolling adhesion and homing on high endothelial venules (HEV)[19,20]. In HIV-1-infected individuals, the number of CD62L+ central memory T cells declines as the disease progresses, resulting in dysfunctional immune responses[21,22]. Despite the apparent clinical association, the molecular mechanism involving L-selectin in HIV biology is not clear.

Here, we investigated the potential role of L-selectin in HIV-1 infection of T cells. We found that L-selectin, despite its preferential binding to sulfated glycoproteins with sialyl-Lewis x moiety[23,24], recognized gp120-associated glycans, and the binding facilitated the viral adhesion and infection. Unexpectedly, we also found that L-selectin shedding is required for HIV-1 release from infected cells. Current anti-HIV therapies target primarily viral protease, reverse transcriptase, and integrase[25,26]. No compounds target viral release. Our findings reveal new pathways for developing antiretroviral treatments targeted at metalloproteinases critical for HIV release.

## Results

**L-selectin binds to HIV-1 gp120 in solution and on cells.** HIV-1 envelope gp120 is highly decorated with N-linked glycans[27,28]. While L-selectin is known to recognize HEV-associated O-linked glycans to facilitate lymphocyte rolling adhesion and homing[29,30], it can also bind to certain N-linked glycans in the absence of O-linked glycosylation[23,24]. To determine if L-selectin recognized glycosylated gp120, we performed surface plasmon resonance (SPR) binding experiments using recombinant gp120 and soluble human L-selectin (CD62L-Fc). Surprisingly, recombinant gp120 from both R5- (HIV-1$_{BAL}$) and X4- (HIV-1$_{SF33}$) strains bound to the soluble L-selectin with 50–300 nM affinities (Fig. 1a, Supplementary Figures 1A, 1B). Removal of the N-linked glycans with peptide N-glycosidase F (PNGase F) reduced the binding of both gp120 to DC-SIGN, a C-type lectin receptor known to recognize N-linked gp120 glycans (Fig. 1b). Likewise, the deglycosylation also abolished gp120 binding to L-selectin (Fig. 1b, Supplementary Figure 1C), suggesting the involvement of N-linked gp120 glycans in L-selectin binding. The carbohydrate specificity of the L-selectin and gp120 binding was further examined using an enzyme-linked immunosorbent assay (ELISA) in the presence of EDTA and various competing carbohydrates (Fig. 1c). EDTA and other known L-selectin ligands, such as heparin, fucoidan and sialyl-Lewis x significantly inhibited gp120

binding, consistent with the involvement of the receptor C-type lectin domain in the viral glycan recognition. In addition, sialyllactose but not N-acetylglucosamine or lactose blocked gp120 binding, supporting the involvement of sialyllated N-linked glycans in L-selectin binding[11]. To investigate if gp120 binds to cell-surface-expressed L-selectin, we conjugated gp120 to fluorescent Qdots and detected their binding to L-selectin-transfected Hela cells (Supplementary Figures 2A, 2C). The gp120-Qdots exhibited specific binding to plate-immobilized recombinant CD4 and L-selectin (Supplementary Figure 1D), and they bound significantly better to L-selectin-transfected than untransfected HeLa cells (Fig. 1d, Supplementary Figures 2B and 2D). As L-selectin is expressed on CD4+ T cells and partially colocalized with CD4 (Supplementary Figures 2E–2H), we then incubated gp120-Qdots with CD4+ human peripheral blood mononuclear cells (PBMC) in the presence of CD4 or L-selectin blocking antibodies. As expected, the gp120 binding was reduced by the CD4 blocking antibody. Importantly, the gp120 binding was also reduced by the L-selectin blocking antibody, DREG 56, and the binding was further reduced by the combination of CD4 and L-selectin blocking antibodies (Fig. 1e). To further address if L-selectin binds to gp120 on HIV-1 virions, we performed virus capture experiments using either plate-bound soluble L-selectin or L-selectin-transfected 293T cells to capture replication-competent HIV-1$_{BAL}$ virus. Both plate-bound L-selectin and CD4 captured significantly more HIV-1$_{BAL}$ than the controls, as measured by p24 ELISA (Fig. 1f). Similarly, more viruses were captured by L-selectin transfected than the untransfected 293T cells (Fig. 1g). As L-selectin is expressed on both CD4+ and CD8+ T cells, we then examined the binding of HIV-1$_{BAL}$ virus to CD62L-sorted CD8+ T cells from PBMC to avoid the involvement of CD4 (Supplementary Figure 3A). Consistently, the sorted CD62L+ CD8+ T cells captured significantly more virus than CD62L− CD8+ T cells (Fig. 1h). Collectively, these results demonstrated L-selectin recognition of HIV-1 gp120.

**L-selectin facilitates HIV viral adhesion and infection.** To explore if L-selectin binding affected HIV-1 infection, we transfected human L-selectin cDNA into REV-CEM, a CD4+ T-cell line susceptible to X4-tropic HIV infections[31]. Stable transfectants (CD62L-CEM # 2 and #25) expressed significantly more L-selectin but similar levels of CD4, CCR5, and CXCR4 compared to the parental cell line (Supplementary Figure 4). Importantly, the infections of CD62L-CEM #2 and #25 by a replication-competent X4-tropic HIV$_{LAI}$ were 3-to-4-fold higher than that of the parental CEM cells as measured by intracellular viral capsid p24 staining (Fig. 2a), suggesting that L-selectin facilitated HIV-1 infection. Since the parental cells expressed low levels of L-selectin (Supplementary Figure 4), we further knocked-down CD62L expression using CRISPR/Cas 9 method[32]. Two clones (CD62L KD #8 and #32) lost L-selectin expressions while the third one (CD62L KD #3) retained the parental level of CD62L expression (Supplementary Figure 4). Despite having comparable CD4 and chemokine receptor expressions, CD62L KD #8 and #32, but not clone #3 were more resistant to HIV$_{LAI}$ infections than their parental CEM cells (Fig. 2a). Finally, HIV$_{LAI}$ infection of CD62L KD #8 was reverted back to that of the parental CEM cells by the addition of polybrene (Fig. 2b), a compound known to increase viral-host adhesion by reducing charge repulsion[33], suggesting L-selectin-facilitated viral adhesion.

To examine if L-selectin affected HIV-1 infection of primary CD4+ T cells, we carried out HIV-1$_{BAL}$ infection of CD8-depleted PBMC in the presence of EDTA, a calcium chelator that inhibited L-selectin binding to gp120 (Fig. 1c). While EDTA did not inhibit HIV-1$_{BAL}$ infection of human macrophages[11], it significantly

reduced HIV-1$_{BAL}$ infection of the PBMC (Fig. 2c, Supplementary Figure 3B). Further, DREG-56 significantly inhibited HIV-1$_{BAL}$ infection of PBMC, particularly at lower viral doses (Fig. 2d), suggesting that L-selectin also facilitated HIV-1 infection of PBMC. To distinguish the contribution of L-selectin to viral entry and replication, we examined viral entry using JRFL- (R5-tropic) and SF33- (X4-tropic) pseudotyped replication deficient viruses. Both JRFL and SF33 infections of PBMC were diminished by the L-selectin blocking antibody (Fig. 3a), and the inhibition was evident over a broad range of viral doses (Fig. 3b, c), supporting the involvement of L-selectin in viral adhesion and entry. To further address the contribution of viral glycans to HIV-1 entry, we produced JRFL and SF33 viruses as well as recombinant SF33 gp120 in HEK 293T or 293S GnTI$^-$ cells, the latter is deficient in mature complex N-glycan productions[34]. Both complex glycan-deficient JRFL and SF33 viruses infected PBMC less than their glycan sufficient counterparts (Fig. 3d), consistent with the

solution binding result showing a reduced L-selectin binding by the complex glycan-deficient gp120$_{SF33}$ (Supplementary Figure 1E). In addition, removal of the gp120 glycans with PNGase F significantly reduced HIV-1$_{BAL}$ infection compared to the mock-treatment (Fig. 3e). The importance of viral glycans in HIV-1 infection is further supported by recent findings that cells from glycosylation-deficient individuals had reduced susceptibility to HIV-1 infection[35].

**HIV-1 infection results in L-selectin shedding**. HIV-1 infection is known to downregulate CD4 expression through internalization[36,37]. Consistent with published data, HIV-1$_{BAL}$ infection resulted in a significant loss of CD4 expression in p24$^+$ but not in p24$^-$ or uninfected T cells on day 6 post-infection (p.i.) and the loss of CD4 was further extended to day 11 p.i. (Fig. 4a−c, Supplementary Figure 3B). Similarly, downregulation in L-selectin expression was also observed in infected,

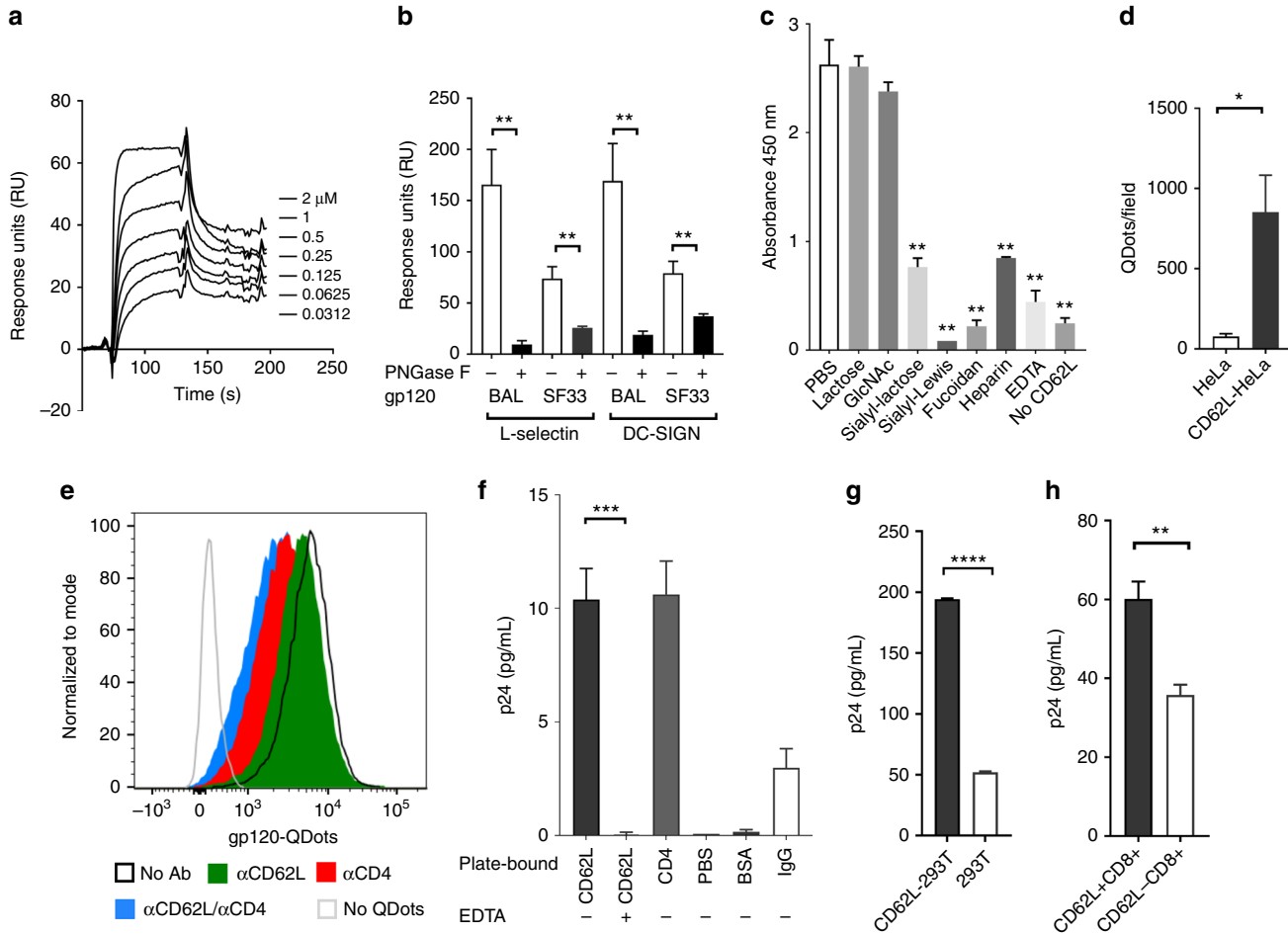

**Fig. 1** L-selectin binds to HIV envelope gp120. **a** Blank subtracted sensorgrams of serially diluted recombinant gp120$_{BAL}$ binding to immobilized CD62L-Fc. **b** Binding of glycosylated and deglycosylated gp120 at ~2 µM concentration to immobilized CD62L-Fc and DCSIGN-Fc. The BIAcore bindings were performed as triplicates using PNGase F treated and untreated gp120s at equal concentrations. Deglycosylated gp120$_{BAL}$ and gp120$_{SF33}$ bound significantly less to both CD62L and DC-SIGN compared to their glycosylated gp120. **c** ELISA binding of CD62L-Fc to immobilized gp120 in the presence of 10 mM lactose, GlcNAc (N-acetyl-ᴅ-glucosamine), sialyl-lactose, sialyl-Lewis x (sLe$^x$), fucoidan, heparin, and EDTA. sLe$^x$, fucoidan, and heparin are known ligands of CD62L and inhibited gp120 binding. The p values are calculated with respect to PBS control. **d** Number of gp120-QDots bound to untransfected or L-selectin-transfected HeLa cells and counted using TIRF microscopy. **e** Flow cytometry analysis of gp120-QDots binding to PBMC in the presence and absence of anti-CD4 (RPA-T4), and anti-CD62L (DREG-56) antibodies. **f** Capture of HIV-1$_{BAL}$ virus (1:1000 dilution) with plate-immobilized 10 µg soluble CD62L in the presence and absence of 5 mM EDTA, CD4, BSA, IgG or blank (PBS control). The captured virus was quantified using p24 ELISA (PerkinElmer). **g** The binding of HIV-1$_{BAL}$ virus (1:1000 dilution) to 10$^6$ CD62L-transfected or untransfected 293T cells. **h** The binding of HIV-1$_{BAL}$ virus to 10$^6$ sorted CD62L$^+$ or CD62L$^-$ CD8$^+$ T cells. The error bars indicate standard deviations from the means. The p values of all figures are calculated using unpaired Student's t test with *$p \leq 0.05$, **$p \leq 0.01$, ***$p \leq 0.001$, ****$p \leq 0.0001$. The results included in panels **a−d**, **f−h** are from at least two independent experiments with all data included

but not uninfected T cells, on days 6 and 11 p.i. (Fig. 4a−c). Indeed, the most significant change in the cell populations associated with HIV-1$_{BAL}$ infection was the loss of the CD4$^+$/CD62L$^+$ cells and the gain of the CD4$^-$/CD62L$^-$ cells (Fig. 4d). These results are consistent with observed down-regulation of L-selectin on T cells from HIV-1-infected individuals naïve to HAART[38]. Comparatively, CCR7 and CD27 expressions were minimally affected in the infected cells (Fig. 4c). L-selectin is known to shed from activated T cells during differentiation of central memory to effector memory T cells to facilitate their migration from lymph nodes to peripheral sites of inflammation[39]. Crosslinking of CD4 with HIV-1 gp120 induced L-selectin shedding on resting CD4$^+$ T cells[40]. To assess if the loss of L-selectin expression in the infected cells resulted from its shedding, we measured the soluble L-selectin levels during HIV-1$_{BAL}$ infection. The concentration of soluble L-selectin steadily accumulated in the infected but not uninfected supernatants (Fig. 4e, Supplementary Figures 5A−C), supporting the shedding of L-selectin during HIV infection. The progressive loss of the selectin expression in p24$^+$ T cells correlated with an increase in

viral infection, suggesting that HIV infection-induced L-selectin shedding (Fig. 4f, Supplementary Figure 5D).

**L-selectin shedding leads to central memory CD4$^+$ T-cell loss.** HIV-1 preferentially infects central memory CD4$^+$ T cells[14,16], and reduced infections of T$_{CM}$ are associated with HIV/SIV non-progressors[41,42]. Further, the ability to maintain central memory T cells correlates with spontaneous HIV-1 control, and the ability to reconstitute T$_{CM}$ under HAART indicates a successful response to antiviral treatments[22,43]. As a marker of central memory CD4$^+$ T cells, L-selectin's role in promoting HIV adhesion is consistent with T$_{CM}$ being the preferred target for the viral infection. Indeed, approximately 50% of HIV-1$_{BAL}$-infected (p24$^+$) T cells were central memory CD45RO$^+$/CD27$^+$/CD62L$^+$ T cells on day 6 post-infection (p.i.), and 10–20% of infected cells were effector memory (T$_{EM}$) CD45RO$^+$/ CD27$^-$/CD62L$^{+/-}$ cells (Fig. 5a, Fig. Supplementary Figure 5E). Interestingly, the infected p24$^+$ T cells contained significantly more transitional memory (T$_{TM}$) cells (CD45RO$^+$/

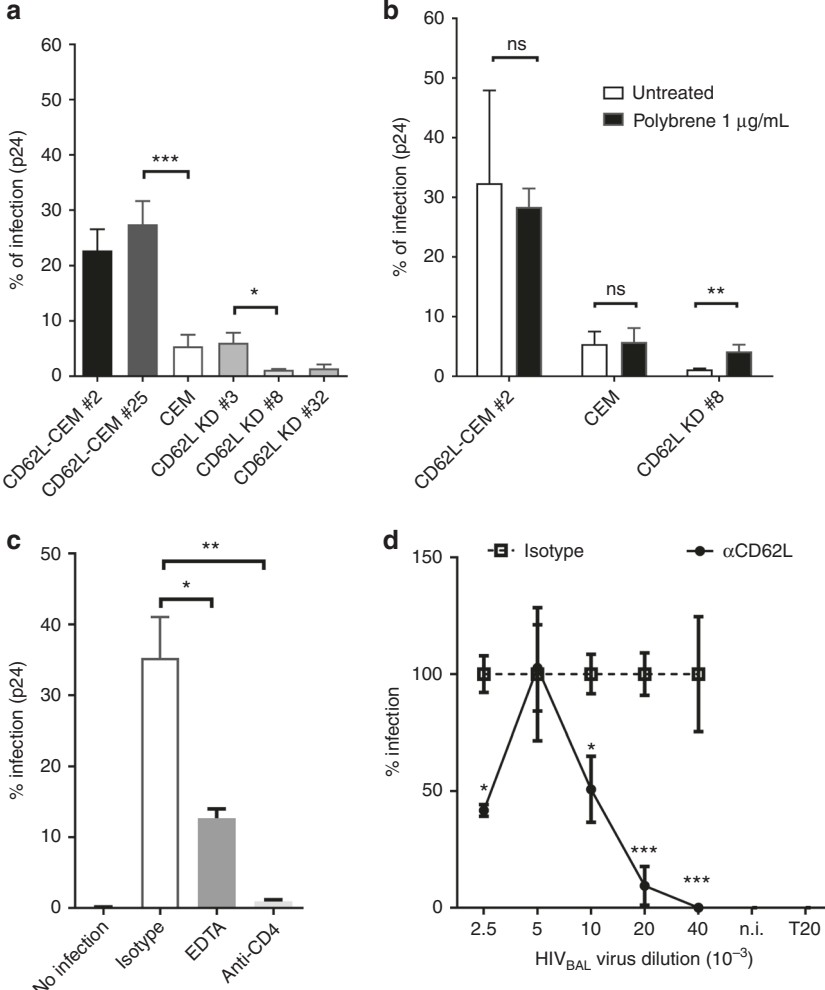

**Fig. 2** L-selectin facilitates HIV-1 infection of CD4$^+$ T cells. **a** HIV infection of CD62L-transfected clones (CD62L-CEM #2 and #25) or untransfected Rev-CEM cells, as well as CD62L knockdown clones, CD62L KD #3, #8, and #32. CD62L KD #8 and #32 lost CD62L expression whereas #3 retained parental CD62L expression (Supplementary Figure 4). **b** HIV infection of CEM cell lines in the presence and absence of 1 µg/mL polybrene. **c** HIV-1$_{BAL}$ infection of PBMC in the presence of EDTA or anti-CD4 (RPA-T4). The infection was measured by intracellular p24$^+$ staining. **d** Dose-dependent HIV-1$_{BAL}$ infection of PBMC in the presence of CD62L blocking antibody DREG-56 or isotype controls. Infections were measured as copy numbers of viral DNA by real-time PCR and displayed as % relative to the isotype controls. The results are from at least two independent experiments and statistics are performed without data rejections

CD27$^+$/CD62L$^-$) than the p24$^-$ or uninfected controls on day 6 p.i. (Fig. 5a, b), and the T$_{TM}$ population increased further on day 11 p.i. (Fig. 5a, b). This increase in infected T$_{TM}$ cells was mirrored by a decrease in the T$_{CM}$ numbers compared to the uninfected controls, suggesting that the viral infection resulted in the memory CD4$^+$ T cells transitioning from T$_{CM}$ to T$_{TM}$. In comparison, a smaller increase in the number of CD45RO$^+$/CD27$^-$ T$_{EM}$ cells was observed between the p24$^+$ and p24$^-$ populations (Fig. 5b). Thus, HIV-induced CD62L shedding likely results in the loss of central memory CD4$^+$ T cells in infected individuals.

Previous studies showed that effector memory CD4$^+$ T cells from HIV-1-infected individuals are dysfunctional in cytokine productions[44,45]. To further investigate the link between HIV infection and the loss of effector memory function, we stimulated HIV-1$_{BAL}$-infected PBMC to produce cytokines and measured their production of IFN-γ, TNF-α, and MIP-1β by intracellular staining using flow cytometry. As expected, T$_{CM}$ consistently produced less cytokines than T$_{EM}$ cells (Fig. 5c)[46]. However, HIV-1$_{BAL}$ infection did not result in reduced IFN-γ production compared to uninfected cells (Fig. 5c), regardless of whether it was from central or effector memory T cells. Further, p24$^+$ T$_{CM}$ and T$_{EM}$ cells consistently secreted more IFN-γ, TNF-α, and MIP-1β than their p24$^-$ populations (Fig. 5d), suggesting that acutely infected cells are not dysfunctional in cytokine productions. We speculate that the previously reported T$_{EM}$ dysfunction may result from grouping of infected T$_{TM}$ (i.e. the CD62L shed T$_{CM}$) as T$_{EM}$ cells, based on L-selectin expression. Since T$_{CM}$ produce less cytokines than T$_{EM}$ cells, this would result in an apparent decrease in cytokine production in T$_{EM}$ cells. This does not exclude the possibility of dysfunction related to memory T cells exhaustion under persistent chronic HIV infections.

**L-selectin shedding is required for HIV-1 viral release**. To investigate if HIV-induced L-selectin shedding affects viral pathogenesis, we explored the inhibition of L-selectin shedding. While the proteases responsible for HIV-induced selectin shedding on CD4$^+$ T cells remain to be identified, ADAM17 is known to shed L-selectin in response to neutrophil activation and the selectin shedding can be inhibited by Batimastat (BB-94)[47–49]. Indeed, BB-94 significantly inhibited L-selectin shedding on CD62L-transfected, but not knockdown CEM cells (Fig. 6a). To our surprise, the inhibition of L-selectin shedding, instead of enhancing the viral infection, significantly reduced their susceptibility to HIV$_{LAI}$ infection (Fig. 6b). Similarly, BB-94 inhibited both X4$^-$ and R5-type HIV infection of PBMC (Fig. 6c, d, Supplementary Figure 6A). To confirm BB-94 inhibited L-selectin shedding during the viral infection, we detected the change of cell surface L-selectin expression, the presence of soluble L-selectin in the media, and the presence of a cleaved C-terminal L-selectin transmembrane fragment on cell surface during HIV-1$_{BAL}$ infection. The results showed that BB-94 prevented HIV-1$_{BAL}$-induced downregulation of L-selectin on PBMC, inhibited the accumulation of soluble L-selectin, and suppressed the L-selectin cleavage fragment (Fig. 6e−g, Supplementary Figure 6B). The specificity of BB-94 is demonstrated by contrasting with a related matrix metalloproteinase (MMP) inhibitor, dichloromethylene-diphosphonic acid (DMDP). While both BB-94 and DMDP inhibit the enzymatic cleavage of MMP-1, DMDP did not block the downregulation of L-selectin expression nor its cleavage, and had no effect on HIV-1$_{BAL}$ infection of PBMC (Supplementary Figures 6C–F). Consistent with the involvement of ADAM17 in HIV-induced L-selectin shedding, the HIV-1$_{BAL}$ infection of PBMC was significantly inhibited by ADAM17-specific inhibitors

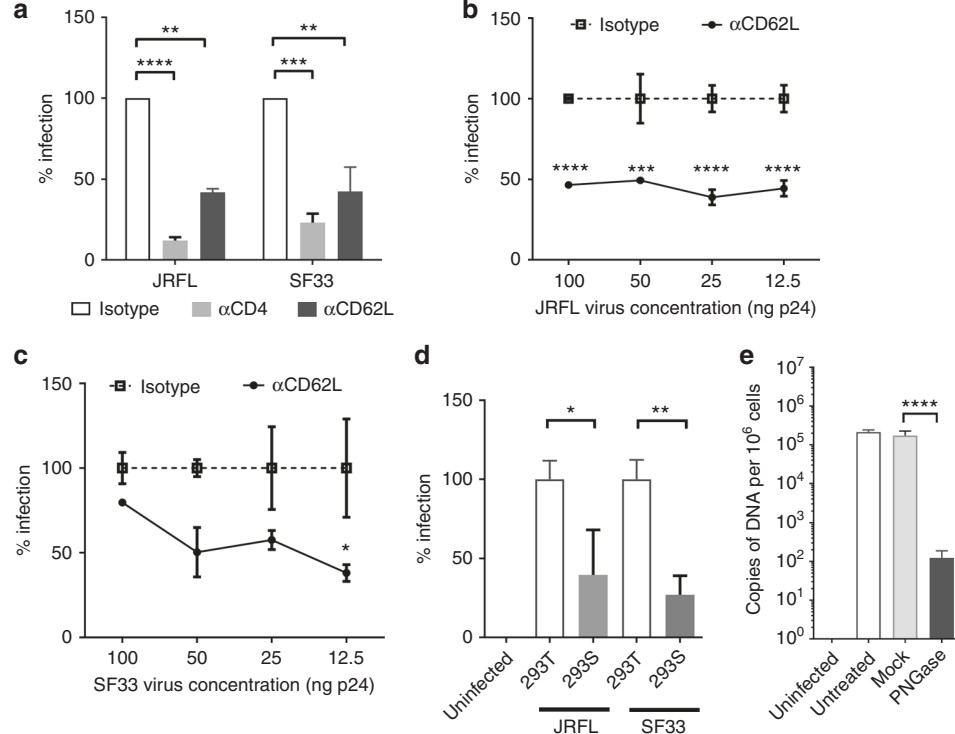

**Fig. 3** L-selectin contributes to HIV-1 adhesion and entry. **a** Infection of PBMC with JRFL and SF33-pseudotyped viruses in the presence of anti-CD62L (DREG-56), anti-CD4 (RPA-T4) blocking antibodies or isotype controls. **b**, **c** Effect of L-selectin blockade in pseudovirus infections. PBMC were infected with serial dilutions of JRFL (**b**), SF33 (**c**) viruses in the presence of DREG-56 or isotype controls. Infections were measured by luciferase activities on day 3 p.i. **d** Infection of PBMC with equal amount of JRFL- and SF33-pseudotyped viruses produced in either 293T or 293S GnTI- cells. **e** The infection of PBMC by PNGase F- or mock-treated HIV-1$_{BAL}$ viruses. The infection level was measured as copies of HIV-1 DNA per 10$^6$ cells by real-time PCR. The results are from two independent experiments with all data included in statistical analysis

TAPI-1 (TNF-α protease inhibitor-1) and TAPI-2 [50] (Fig. 6h). Additionally, the viral infection-induced L-selectin cleavage resulted in an identical sized C-terminal fragment as that induced by PMA treatment, known to induce ADAM17 cleavage of L-selectin (Fig. 6g)[51].

To examine if L-selectin shedding affected HIV entry, we infected PBMC with JRFL and SF33 pseudoviruses in the presence of BB-94. In contrast to HIV-1$_{BAL}$ infection, BB-94 did not affect significantly the entry of JRFL and SF33 (Fig. 6i), suggesting L-selectin shedding affects post entry of the viral infection. Since L-selectin shedding likely occurred late in infection (Fig. 4), we then assessed the effect of BB-94 to HIV release using a trypsin-mediated viral release assay. HIV-1$_{BAL}$-infected cells were treated with trypsin on day 6 p.i. to release cell

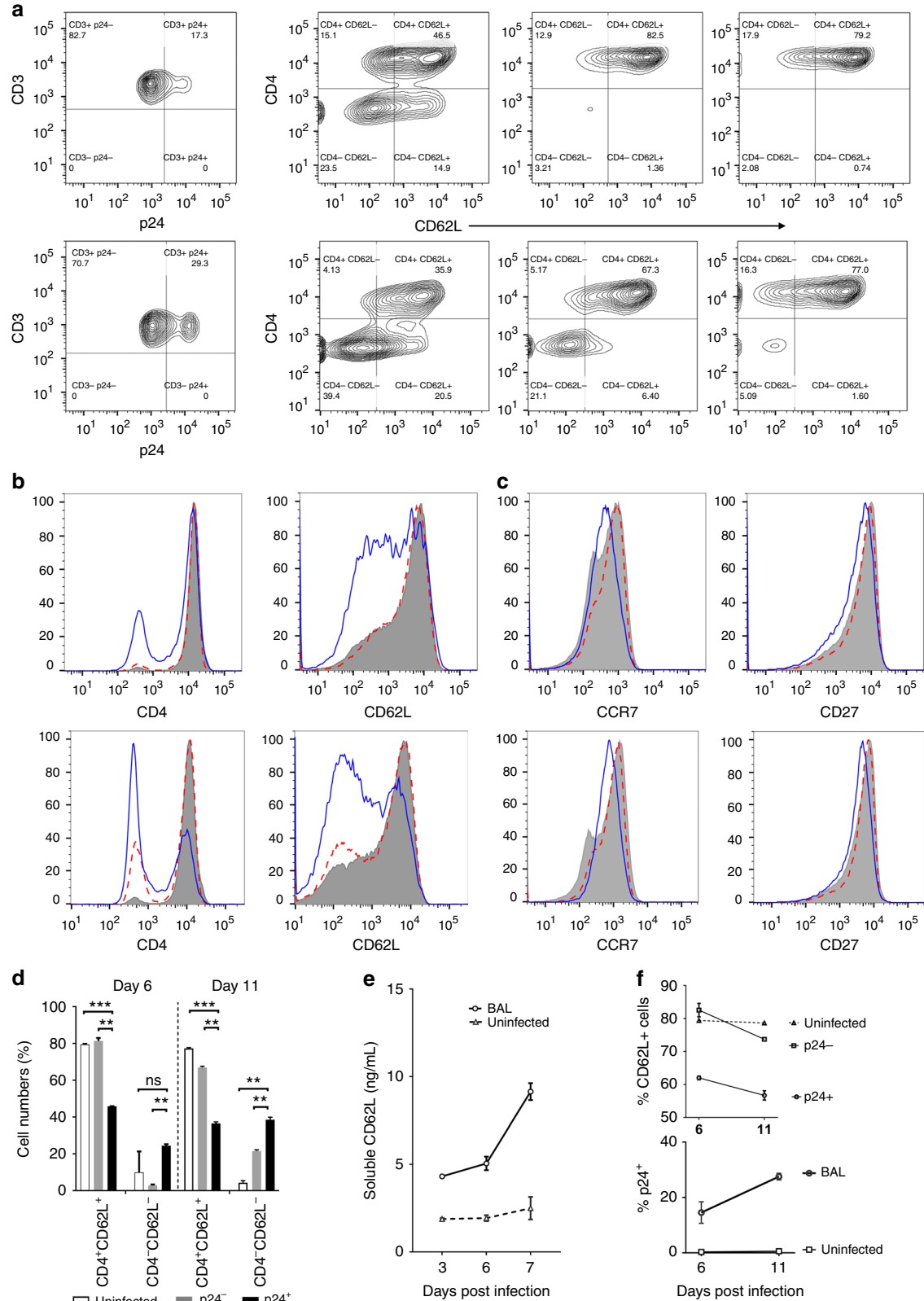

surface-associated viral particles. While BB-94 decreased the amount of virus detected in the supernatant prior to and without trypsin treatment, trypsin treatment resulted in the release of significantly more viral particles in the presence of BB-94 (Fig. 6j), suggesting BB-94 inhibited viral release. To visualize the effect of BB-94 on the viral release, we examined HIV-1$_{BAL}$-infected cells using scanning and transmission electron microscopy (SEM/ TEM). In the absence of BB-94, the majority of budding viruses from HIV-1$_{BAL}$-infected PBMC exhibit typical size of ~100 nm (Fig. 7a, b, e)[52]. In the presence of BB-94, however, more virus-like particles were observed on infected cells, and many of them appeared significantly smaller than the size of budding virions in the absence of BB-94 (Fig. 7c, f, g), suggesting they may be defective viral particles. The retention of budding viruses in the presence of L-selectin shedding inhibitors suggests the selectin shedding is required for viral release. As HIV-1 viruses can infect target cells through cell−cell transfer, we then evaluated if the selectin shedding also affects viral release in cell−cell transfer infections[53,54]. When TZM-BL cells were incubated with HIV-1$_{BAL}$-infected PBMC, the cell−cell transfer-mediated infection of TZM-BL cells was suppressed by BB-94 in all ratios of PBMC to TZM-BL (Fig. 6k). It is worth noting that BB-94 did not significantly affect VSV infections (Supplementary Figure 6G), suggesting L-selectin shedding impacts some but not all viral infections.

To further investigate if L-selectin shedding is required for HIV release in infected individuals, CD4$^+$ T cells from HIV-1-infected viremic and aviremic individuals were stimulated with anti-CD3 to induce viral release. The viremic individuals had viral loads between 1200 and 100,000 copies of HIV-1 RNA/mL, while aviremic individuals were on ART regiments and had undetectable plasma viremia (<40 copies HIV RNA/mL). Consequently, anti-CD3 induced 10–100-fold more virion release from viremic than aviremic CD4$^+$ T cells (Fig. 7h). Importantly, the release of HIV from both viremic and aviremic individuals were profoundly inhibited by BB-94, but not by DMDP (Fig. 7h). When the viral release was examined using CD4$^+$ T cells from six HIV-1-infected individuals, BB-94 inhibited 30−90% of all viral releases (Fig. 7i). Together, these data suggest that L-selectin shedding is also required for HIV-1 release in vivo. While the mechanism regulating the selectin shedding and HIV-1 release remains to be elucidated, it is possible that failure to shed the selectin results in cell surface retention of budding virions through L-selectin binding to gp120. To assess if a budding virion interacts with L-selectin in the presence of BB-94, we performed immuno-TEM on HIV-1$_{LAI}$-infected Rev-CEM to observe budding virions in the presence of immunogold-conjugated anti-CD62L. While gold particles were readily found on infected cells, they were generally not associated with budding virions (Fig. 8a, b). In the presence of BB-94, however, a significant number of gold particles were associated with budding virions at the budding focal point (Fig. 8c, d), suggesting the presence of selectin tethering (Fig. 9).

## Discussion

L-selectin (CD62L) provides rolling adhesion for lymphocyte extravasation to secondary lymph nodes and sites of inflammation[20]. The binding of HIV-1 glycans to L-selectin can be viewed similarly as viral rolling adhesion on CD4$^+$ T cells to facilitate the binding to CD4 and other coreceptors, and thus to enhance viral infections. Biochemically, the recognition of gp120 represents a novel function for L-selectin in addition to the binding of O-linked glycans. While L-selectin prefers sialyl-Lewis x type of O-linked glycans, its binding to gp120 is likely the result of engaging multiple low affinity N-linked glycans on heavily glycosylated envelope proteins. The carbohydrate promiscuity of L-selectin is supported by a recent crystal structure of L-selectin binding to a mannose moiety on an N-linked glycosylation[55]. Importantly, we demonstrated the involvement of L-selectin in HIV-1 viral entry and infection. Thus, HIV viral envelope glycans not only shield the virus from host immune detection, but also provide necessary adhesion to macrophages and CD4$^+$ T cells for viral entry and infection[10,11]. It is worth noting that since the viral glycan-mediated L-selectin binding depends primarily on viral envelope glycosylations, other viruses with highly glycosylated envelope proteins may also bind L-selectin to facilitate their adhesion and entry.

Early experiments suggested the ability of HIV-1 envelope to induce L-selectin shedding on resting CD4$^+$ T cells[40], we showed here that HIV-1 infection resulted in a progressive loss in L-selectin expression. Further, the downregulation of L-selectin expression was significantly more in p24$^+$ than p24$^-$ populations suggesting that the selectin shedding was induced by productive viral infection rather than gp120 binding. While shedding of L-selectin on HIV-1-infected T cells may deter additional virions from binding to an already infected cell, preventing so called "super-infection", the inhibition of shedding did not significantly affect the viral entry. Instead, it resulted in the retention of virus-like particles on infected cells. Strikingly, many of the budding virus-like particles appeared smaller than fully formed viruses in the presence of BB-94, suggesting they are defective virions.

In summary, we showed here a direct involvement of L-selectin in HIV-1 attachment to CD4$^+$ T cells to facilitate the viral entry. The L-selectin-mediated viral adhesion explains the preferential infection of central memory CD4$^+$ T cells by HIV-1 virus. Upon infection, however, HIV induces L-selectin shedding by ADAM17 family of metalloproteinases from infected T cells to facilitate the viral release (Fig. 9). It remains unknown which viral genes induce L-selectin shedding. While HIV-1 encoded nef was shown important for CD4 downregulation[56], it would be interesting to see if nef, vpu, or other viral encoded proteins are responsible for activating L-selectin shedding. We speculate that the involvement of L-selectin in HIV-1 adhesion and release is not unique to HIV; rather, the underline mechanism can be generalized to all lectin receptors and their involvement in the adhesion and release of any glycosylated envelope viruses. Despite the success of HAART in reducing viral load in HIV-infected individuals, a cure remains

**Fig. 4** HIV-1 infection resulted in L-selectin shedding. **a** PBMC were infected with HIV-1$_{BAL}$ and CD4 and CD62L expression was measured on days 6 and 11 p.i. for p24$^+$, p24$^-$, and uninfected cells. The results are the representative of three individual experiments. **b** Histograms showing the expression of CD4, CD62L on days 6 and 11 p.i. for p24$^+$ (solid blue), p24$^-$ (dotted), and uninfected (shaded) cells. **c** CCR7 and CD27 expression as defined in **b**. **d** Downregulation of CD4 and CD62L in infected PBMC is evident from the decrease in % of CD4$^+$/CD62L$^+$ cells and the increase in CD4$^-$/CD62L$^-$ cells on day 6 and further on day 11 in p24$^+$ compared to p24$^-$ and uninfected populations. **e** The accumulation of soluble L-selectin in the supernatant of infected and uninfected samples was measured by ELISA on days 3, 6, and 7 post HIV-1$_{BAL}$ infection of PBMC (Supplementary Figure 5A−C). **f** HIV-1 infection of PBMC induced downregulation of L-selectin expression. The loss of CD62L$^+$ CD4 T cells during HIV-1$_{BAL}$ infection (upper panel) is accompanied by the increase in the viral infection (lower panel) at days 6 and 11 p.i. The results are from at least two independent experiments with all data included in the analysis

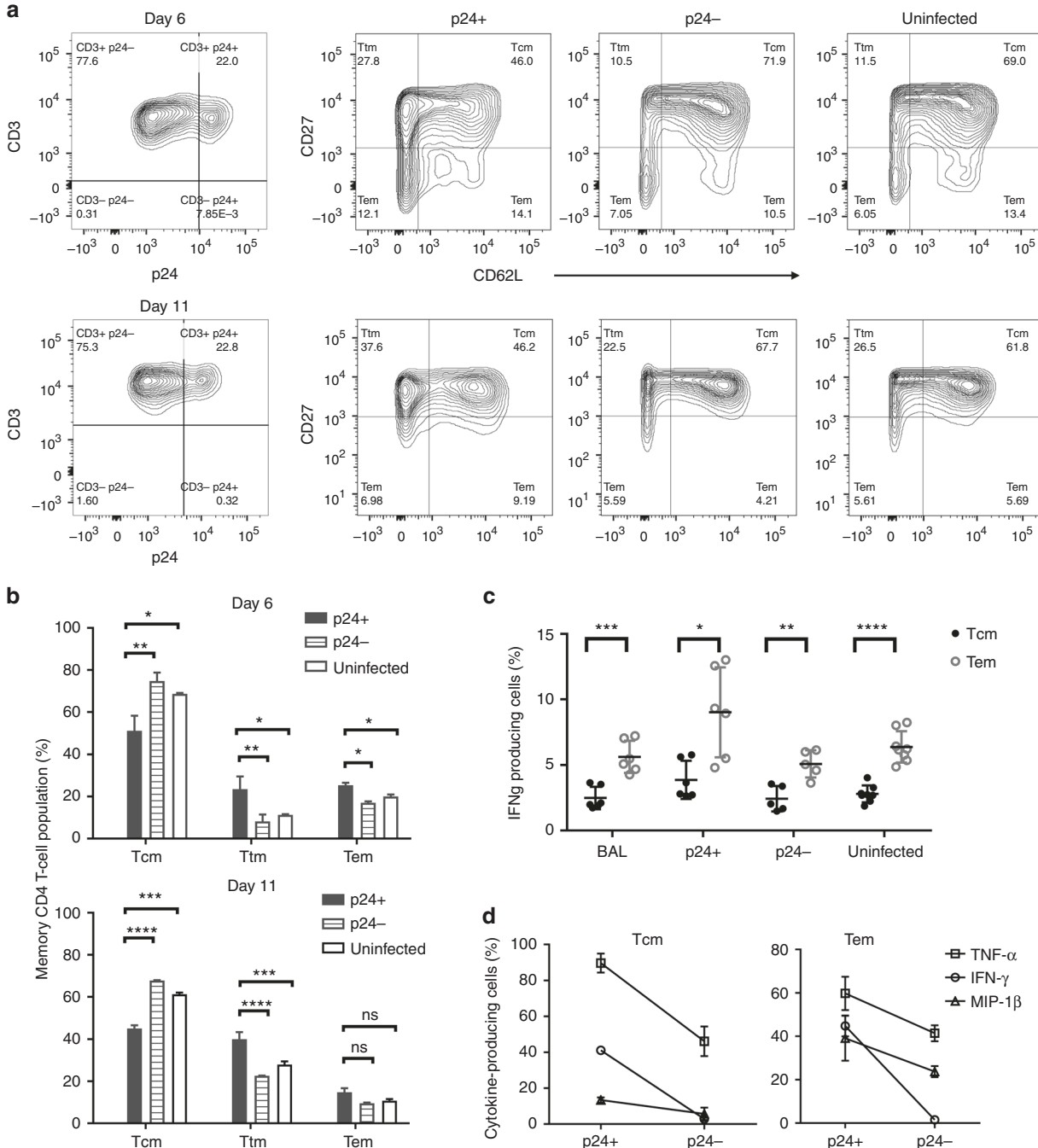

**Fig. 5** HIV-induced L-selectin shedding contributed to a loss of central memory CD4$^+$ T cells. **a** A representative analysis of memory CD3$^+$ T cells (CD45RO$^+$) gated for CD27 vs. CD62L on days 6 and 11 p.i. for p24$^+$, p24$^-$, and uninfected populations. **b** Percentage of memory subset (T$_{CM}$, T$_{TM}$, and T$_{EM}$) populations for p24$^+$, p24$^-$, and uninfected cells on days 6 and 11. The loss of T$_{CM}$ and gain of T$_{TM}$ populations in infected PBMC are significant when comparing p24$^+$ cells with p24$^-$ and uninfected cells. In contrast, % of T$_{EM}$ cells do not differ significantly among p24$^+$, p24$^-$, and uninfected cells. **c** Percentage of interferon-γ-expressing T$_{CM}$ and T$_{EM}$ cells in overall HIV$_{BAL}$ infected and uninfected PBMC as well as p24$^+$ and p24$^-$ populations on day 6 p.i. **d** IFN-γ-, TNF-α-, and MIP-1β-expressing (%) T$_{CM}$ (left panel) and T$_{EM}$ (right panel) cells in p24$^+$ and p24$^-$ populations on day 6 p.i. The results are from at least two independent experiments with all data included in the analysis

elusive in the absence of an effective vaccine. The success of HAART in suppressing plasma HIV-1 viremia has brought renewed focus on finding and eliminating latently infected viral reservoirs through their reactivation followed by elimination with HAART therapy[57–62]. Our finding that L-selectin shedding is required for HIV release either in experimental infections or from patient-derived CD4$^+$ T cells reveals a novel pathway to suppress both active and latent viral release. While the active sheddase on

infected CD4+ T cells remain to be identified, small molecular inhibitors of metalloproteinases can be explored as a new class of antiviral compounds targeted at HIV release.

## Methods

**Reagents.** Unless otherwise specified, all reagents and chemicals were purchased from Sigma-Aldrich Co. (St. Louis, MO). Recombinant protein was purchased from R&D Systems, Inc. (Minneapolis, MN). Blocking antibody against CD62L

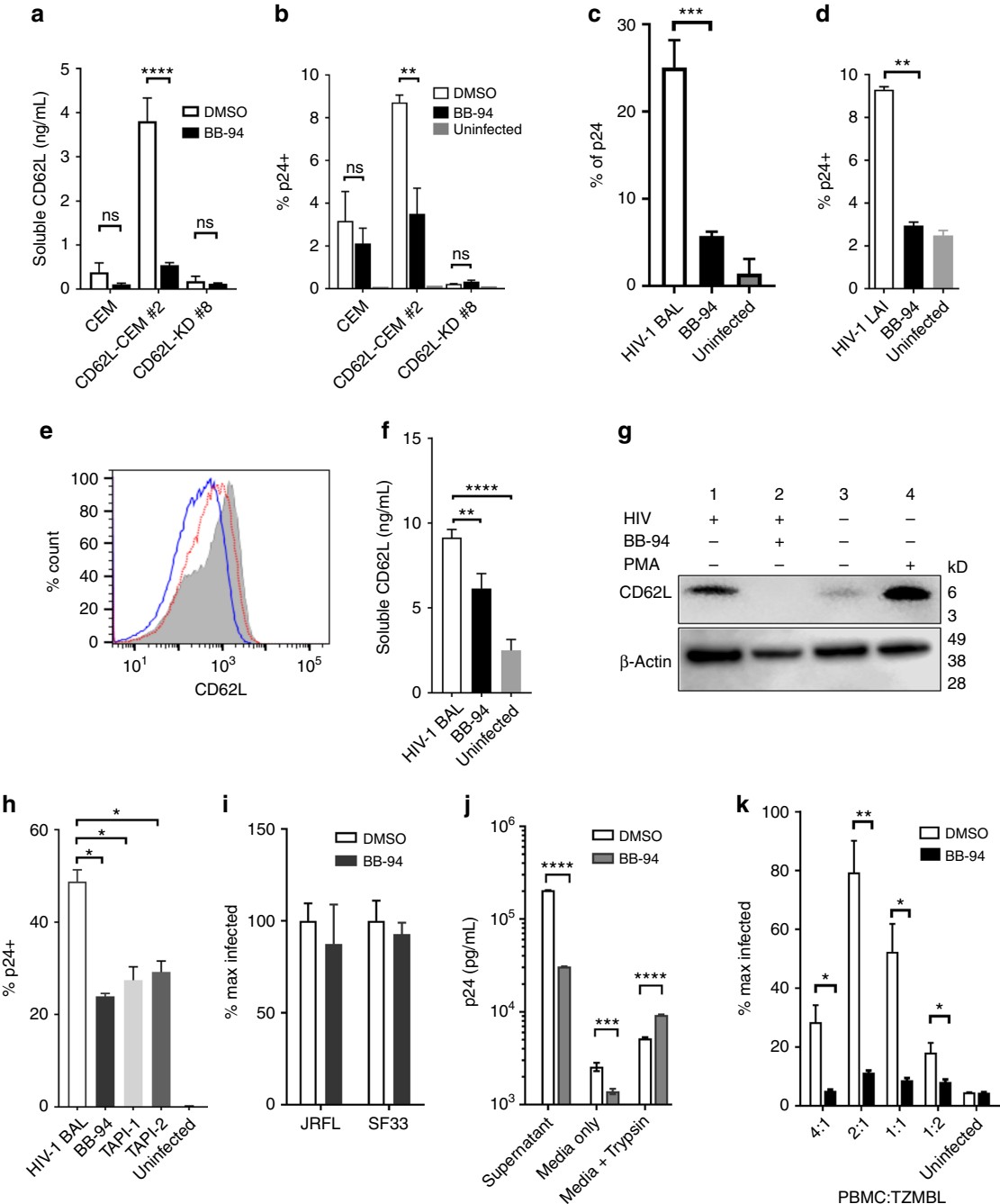

**Fig. 6** L-selectin shedding is required for HIV-1 release. **a** BB-94 inhibition of soluble CD62L shedding from parental CEM as well as CD62L-transfected clone #2 and knockdown #8. The CD62L expression on various CEM cells are shown in Supplementary Figure 4. **b** The inhibition of X4-tropic HIV-1$_{LAI}$ infection of CEM cells by BB-94. **c**, **d** The inhibition of R5-tropic HIV-1$_{BAL}$ (**c**) or X4-tropic HIV-1$_{LAI}$ (**d**) infections of PBMC by BB-94 on day 6 p.i. **e** The expression of CD62L during HIV-1$_{BAL}$ infection (**c**) from uninfected (gray area) and HIV-1$_{BAL}$ infected PBMC in the absence (blue lines) or presence of 100 μM BB-94 (red). **f** Shedding of L-selectin in the infected and uninfected supernatants during HIV-1$_{BAL}$ infection of PBMC (panel c) in the presence and absence of BB-94. **g** The western blot analysis of a cleaved 6kD C-terminal L-selectin fragment in the presence and absence of BB-94 during HIV-1$_{BAL}$ infection (**c**) (**g**). Uninfected PBMC were treated with PMA for 30 min to induce L-selectin shedding. Lanes 1 and 2 are cell lysates from the infected samples in the presence of DMSO or BB-94. Lanes 3 and 4 are cell lysates from uninfected samples in the absence or presence of PMA. β-actin (lower panel) is used as loading control. **h** The effect of ADAM17-specific inhibitors TAPI-1, TAPI-2 on HIV-1$_{BAL}$ infections of PBMC. **i** The infection of PBMC by JRFL- and SF33-pseudotyped viruses in the presence of BB-94 or DMSO. Luciferase activity was measured at day 3 p.i. **j** Trypsin-mediated viral release assay. The release of viral particles upon trypsin digestion from day 6 of infected PBMC in the presence and absence of BB-94 or DMSO was measured by p24 ELISA. **k** BB-94 inhibition of cell–cell transfer-mediated HIV-1$_{BAL}$ infections. TZM-BL cells were infected through coincubation with titrating amount of infected PBMC in the presence of BB-94 or DMSO. The infection of TZM-BL cells was measured by luciferase activity 48–60 h p.i. The results are from at least two independent experiments with all data included in the analysis

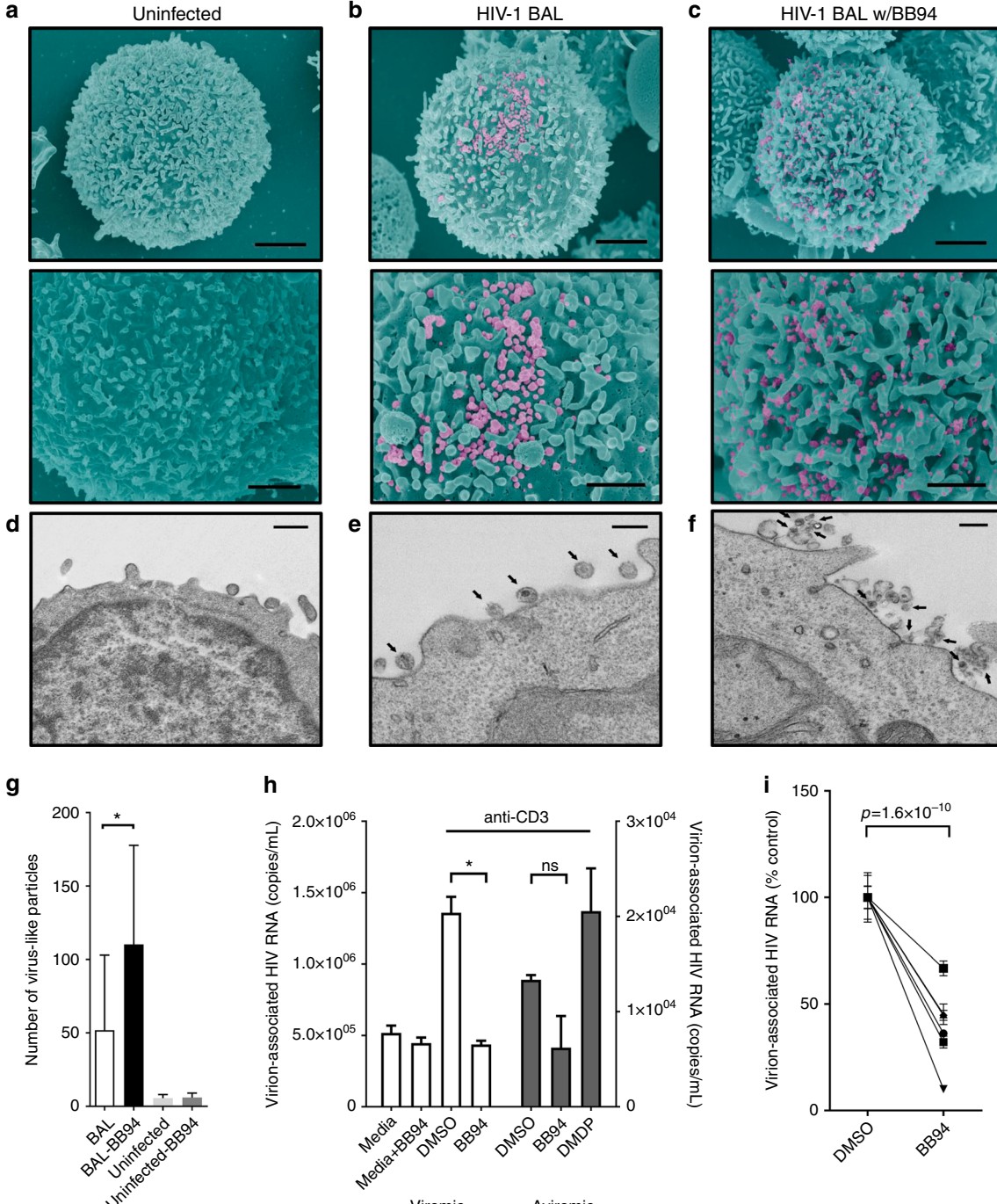

**Fig. 7** Effect of BB-94 on HIV-1 release. **a–f** SEM (**a–c**) and TEM (**d–f**) images of uninfected (**a, d**) and HIV$_{BAL}$-infected (**b, c, e, f**) CD4 T cells on day 6 p.i. in the absence (**b, e**) or presence (**c, f**) of BB-94. Lower panel (**a–c**) is ~×2 higher magnification view of top panel. SEM image scale bars are 2 µm (top panel) and 1 µm (bottom panel). TEM image scale bars are 600 nm (**d**) and 200 nm (**e, f**). The images are representatives from at least two independent experiments. There are ~170 virus particles on the infected cell in the absence of BB-94 and a majority of them display typical size of 100–150 nm (**b, e**). In contrast, ~220 particles are found on an infected cell in the presence of BB-94 with majority of them exhibiting <80 nm size (**c, f**). **g** Average number of virus-like particles observed on HIV-budding cells in the presence and absence of BB-94 were quantified in 5–10 TEM images from two independent experiments. Significantly more virus-like particles were found on infected PBMC in the presence of BB-94. **h** Stimulated virion release from CD4+ T cells of HIV-1-infected individuals. CD4+ T cells from both viremic (left) and aviremic (right) individuals were activated with anti-CD3 antibody or media in the presence and absence of BB-94, DMDP or DMSO. **i** Paired DMSO and BB-94 treatments as in **g** from multiple HIV-1-infected individuals. The results are from at least two independent experiments with all data included in the analysis

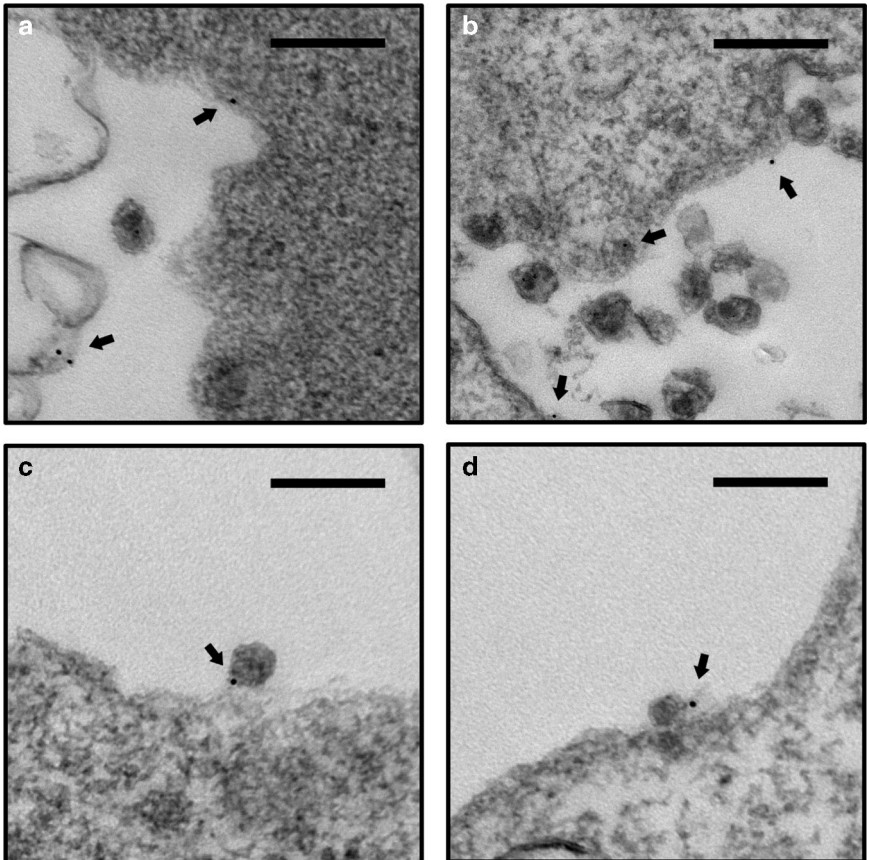

**Fig. 8** Immunogold-labeled budding HIV viruses. Examples of TEM images of HIV-1$_{LAI}$-infected Rev-CEM cells labeled with 10 nm gold particle conjugated anti-CD62L (FMC46) in the absence (**a**, **b**) and presence (**c**, **d**) of BB-94. Arrows point to the gold-labeled CD62L. Gold particles were generally visible on infected T cells near but not associated with HIV budding virions (**a**, **b**) in the absence of BB-94. In the presence of BB-94, however, a significant number of budding virions are observed directly associated with gold particles, suggesting the tethering of the budding virions by L-selectin (**c**, **d**). The images are representatives of at least five similar images in each category. Scale bars are 200 nm

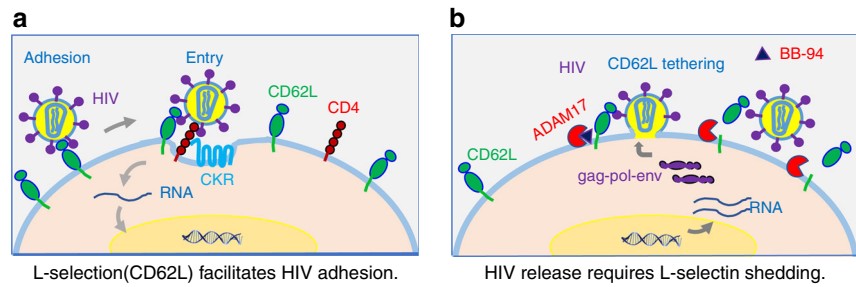

**Fig. 9** Schematic diagrams showing the involvement of L-selectin in HIV-1 infection of CD4$^+$ T cells. **a** L-selectin mediated viral adhesion. **b** The viral release requires the seletin shedding

(DREG-56) was harvested from hybridoma cells in serum-free media from Invitrogen (Carlsbad, CA) and purified using a Protein A column or purchased from eBioscience (San Diego, CA). Unlabeled mouse antihuman CD4 monoclonal antibody (RPA-T4), CD28 were obtained from eBioscience (San Diego, CA). Antihuman CD3 antibody (OKT3) was kindly provided by Dr. Gilliland of Johnson & Johnson. Fluorescently labeled antibodies for flow cytometry against CD14, CXCR4, CCR5, CD8, CD4, CD3, CD62L, CD27, CD45RO, CCR7, IFNγ and their isotype controls (IgG1, IgG2A, IgG2B) were obtained from BD Biosciences (San Jose, CA), BioLegend (San Diego, CA) or eBioscience (San Diego, CA). Alexa-647-labeled antibodies used for confocal microscopy were obtained from BioLegend (San Diego, CA). HIV-1 core antigen antibody (KC57-FITC) for intracellular p24 staining was purchased from Beckman Coulter, Inc. (Miami, FL). Interleukin 2 (IL-2) was obtained from Peprotech Inc. (Rocky Hill, NJ). Polyacrylamide (PAA)-conjugated model carbohydrates were obtained from Glycotech, Inc. (Rockville, MD). All other carbohydrates were purchased from Carbosynth Ltd. (Compton, Berkshire, UK). Recombinant gp120 proteins were expressed either using stably transfected CHO cells or transiently transfected 293T and 293S GnTI⁻ cells in monomeric forms[63]. The Luciferase Assay System was purchased from Promega Corporation (Madison, WI). HIV-1 p24 ELISA kit was obtained from PerkinElmer Life Sciences, Inc. (Waltham, MA). Ficoll-Paque was purchased from GE Healthcare Life Sciences (Pittsburgh, PA). For FACS analysis, recombinant gp120 proteins were labeled with biotin using a biotinylation kit from Pierce Biotechnology (Rockford, IL). RPMI, penicillin/streptomycin, fetal bovine serum (FBS), HEPES, and Versene were purchased from Invitrogen Corporation (Carlsbad, CA). The metalloproteinase inhibitor, BB94 (Batimastat), TAPI-1 (TNFα protease inhibitor-1), and TAPI-2 were purchased from Santa Cruz Biotechnology.

**Activation of peripheral blood mononuclear cells**. Peripheral blood mononuclear cells were isolated by Ficoll-Paque gradient from randomly selected non-identified healthy donors. The use of human PBMC is approved by the Department of Transfusion Medicine at the Clinical Center of National Institutes of Health. The isolated PBMC were distributed at 3×10⁶/mL in 12-well plates with RPMI

supplemented with 10% FBS, 1% penicillin/streptomycin and 20 U/mL IL-2, activated with 1–2 µg/mL anti-CD3 antibody Okt3 for 48 h. CD8+ cell depletion was completed using the StemCell (Vancouver, BC, Canada) EasySep™ Human CD8 Positive Selection Kit prior to infection. Total cell count and viability determinations were assessed with the Guava Personal Cell Analysis System (Millipore) or the Luna FL Dual Fluorescence Cell Counter (Logos Biosystems); all assays were performed with a cellular viability greater than 90%. All data presented in this manuscript were results of individual typical experiments derived from at least two independent experiments.

**Stable CD62L-transfected HeLa, 293T, and Rev-CEM cells**. HeLa cells were cultured in DMEM/F12 medium supplemented with 5% FBS. Neomycin-resistant pEX02-CD62L vector (Genecopeia Inc., Rockville MD) containing coding regions for human CD62L were transfected into HeLa cells using Lipofectamine 2000 from Invitrogen (Carlsbad, CA), and cultured at 37 °C and 5% $CO_2$ in DMEM/F12 supplemented with 5% FBS, 1% penicillin/streptomycin, and 300 µg/mL G418. G418-resistant cells were sorted on an FACS Aria II (BD Biosciences) for CD62L expression. The sorted populations were expanded in the same growth media. To reduce cleavage of selectins from the cell surface, sorted and unsorted transfected cells were grown in Nunc™ UpCell® six-well plates from Thermo Fisher Scientific (Waltham, MA) and released after incubating at room temperature in Versene and gentle pipetting. Rev-CEM cells[31], obtained from the NIH AIDS Research and Reference Reagent Program (https://www.aidsreagent.org/), were transfected with 2 µg of full-length human L-selectin cDNA in pEX02 vector using Nucleofector® Kit V (Lonza, Walkersville, MD). Two days later, cells were selected with 400 µg/mL of G418 and continue to culture for 2 weeks, and then sorted for CD62L expression. Individual clones were screened for CD62L expressions and high expression clones were expanded for infection experiments. CD62L knockout CEM cell lines were established similarly, except SELL CRISPR guide RNA 6 vector was transfected into the parental Rev-CEM cells and cells were selected with 1 µg/mL of puromycin (GeneScript, Piscataway, NJ)[32]. Resistant cells were sorted for the lack of CD62L expression and maintained in RPMI-1640 media supplemented with 10% FBS, 1% penicillin/streptomycin, and 1 µg/ml puromycin.

**Preparation of the pseudotyped HIV viruses**. The HIV viral vector, pNL4-3.Luc.R-E-, which contains the firefly luciferase gene inserted into the NL4-3 HIV nef gene and frameshift mutations to render it E-, was used to generate all pseudotyped viruses[64]. In brief, the expression vectors for pNL4-3.Luc.R-E-, the amphotropic envelope pHEF-VSVG, and the R5-tropic HIV JRFL envelope were obtained through the NIH AIDS Research and Reference Reagent Program. Expression vectors for the X4-tropic SF33 HIV-1 envelope was obtained from M. Martin[65]. Recombinant HIV luciferase viruses were generated by cotransfecting 293T cells with 5 µg of the NL4-3 backbone and either 5 µg of the HIV envelopes or 1.5 µg of the VSV envelope, as previously described[66]. Virus collected in the culture supernatant were quantified by HIV p24 ELISA and adjusted to 1 mg/mL p24. Pseudotyped virus deficient for complex carbohydrates were generated similarly except with a transfected HEK 293S GNTI- cells, obtained from American Type Culture Collection (Manassas, VA). Activated CD8-depleted PBMC (CD8- PBMC) were infected with equal amounts of pseudotyped HIV-1 viruses, as measured by their p24 concentration.

**Replication competent virus**. The R5-tropic BAL strain of HIV-1 (HIV-1$_{BAL}$), propagated using primary human macrophages, was purchased from Advanced Biotechnologies Inc. (Columbia, MD). The X4-tropic HIV-1 LAI was obtained from the NIH AIDS Research and Reference Reagent Program (https://www.aidsreagent.org/). Both viruses were propagated by infecting CD8- PBMC. Day 6 supernatant was harvested and 200 µL aliquots were frozen. Viral titers in TCID$_{50}$ were determined by titrating viral infection in CD8- PBMC and measuring p24 ELISA using PerkinElmer ALLIANCE HIV-I p24 ELISA kit (PerkinElmer Life Sciences, Inc., Waltham, MA).

**BIAcore, ELISA binding, and viral capture experiments**. Surface plasmon resonance measurements were performed using a BIAcore 3000 instrument (GE Healthcare Life Sciences). Recombinant CD62L-Fc, DC-SIGN Fc fusion proteins (R&D Systems, Inc.), or anti-gp120 antibody 2G12 were immobilized onto either CM5 or C1-sensor chips by N-hydrosuccinimide/1-ethyl-3(-3-dimethylamino-propyl) carbodiimide hydrochloride(NHS/EDC) crosslinking in sodium acetate buffer at pH5.0. Human IgG$_1$ was immobilized as control to Fc fusion proteins. Binding assays were performed in HBS-P buffer (10 mM HEPES pH 7.2, 150 mM NaCl, 0.005% P20) (GE Healthcare Life Sciences), supplemented with 0.5–2 mM CaCl$_2$, and 0.5 mg/mL bovine serum albumin (BSA) to reduce nonspecific binding. Recombinant gp120 proteins with varying concentrations between 0.003 and 2 µM, in the HBS-P + Ca buffer, were injected over immobilized receptors. To deglycosylate gp120, 50 µg of gp120 was treated with 100 units PNGaseF (New England Biolabs) under nondenaturing conditions for 3 h at 37 °C following the manufacturer's guide, and then dialyzed overnight in HBS-P binding buffer. The dissociation constants ($K_D$) were determined from kinetic curve fitting using the BIAevaluation software on blank subtracted sensorgrams (GE Healthcare Life Sciences). To detect gp120 and CD62L binding by ELISA, 50 ng of gp120 proteins

were immobilized in individual wells of a 96-well plate for 1 h at room temperature in coating buffer (10 mM Tris [pH7.5] and 2 mM CaCl$_2$), blocked for 1 h using blocking buffer (10 mM Tris [pH7.5], 0.1% Tween20), and washed three times with the same buffer. CD62L-Fc (25 ng) was added to each well in the presence or absence of various inhibitors (10 mM EDTA or 10 mg/mL carbohydrates) together with a goat antihuman IgG-HRP secondary antibody for 1 h at room temperature. The plate was washed five times and readout was colorimetric using a TMB substrate and analyzed on a SpectraMax Plus 384 spectrophotometer (Molecular Devices). All statistics are calculated using unpaired Student's t test.

To perform viral capture ELISA, 96-well plates were coated overnight at room temperature with 10 µg protein per well of CD4, CD62L-Fc, BSA, IgG1 (R&D Systems) or PBS. The wells were washed with ELISA wash buffer (PerkinElmer ALLIANCE HIV-I P24 ELISA), and blocked for 1 h at room temperature with 1% BSA in 1× PBS. HIV-1$_{BAL}$ was added to each well at a 1:1000 dilution of virus for 1 h at 37 °C in the presence or absence of 5 mM EDTA. The wells were gently washed with PBS, and Triton X was added to denature the bound virus. The amount of captured virus was determined using the PerkinElmer p24 ELISA kit. To capture HIV-1$_{BAL}$ by CD62L-transfected 293T cells or CD62L-sorted CD8+ T cells, HIV-1$_{BAL}$ virus (1:1000 dilution) were mixed with 10$^6$ CD62L-transfected or untransfected 293T cells, sorted CD62L+ or CD62L- CD8+ T cells at m.o.i. of ~0.1, incubated for 1 h at 37 °C. The cells were gently washed with PBS, treated with Triton X. The amount of cell-bound virus was determined as above.

**Single-round infection assay**. Stimulated CD8- PBMC were resuspended at 2×10$^6$/mL in culture media. Aliquots of 200 µL (4×10$^5$ cells) were transferred to 96-well plates for incubation in triplicate with anti-CD4 (10 µg/mL), anti-CD62L (30 µg/mL), or isotype (30 µg/mL) antibody at 37 °C for 60 min prior to the addition of virus. Luciferase viruses pseudotyped with envelopes from JRFL (R5-tropic) and SF33 (X4-tropic) HIV-1 and VSV were added to the cells at a concentration of 100 ng/mL HIV p24. The infected CD8- PBMC were then incubated for 72 h, lysed, and assayed for luciferase activity according to the manufacturer's recommendations (Promega Corporation, Madison, WI). Pseudotyped virus from GNTI- cells were added at 100 ng/mL to cells as above.

**Infection with replication-competent viruses**. Rev-CEM cells expressing differing levels of CD62L were infected with 1:100 v/v dilution of HIV-1$_{LAI}$ at 2×10$^5$ cells/mL in a 48-well plate along with uninfected controls at 37 °C with 5.5% $CO_2$. Media was replaced every 3 days post-infection. For infection of PBMC, CD8-depleted PBMC were resuspended at 2×10$^6$/mL in culture media. Cells (1 mL) were incubated with antibodies or inhibitors at 37 °C for 60 min prior to infection. The concentrations used were: 10 µg/mL anti-CD4, 30 µg/mL anti-CD62L, 10 or 30 µg/mL isotype antibody in matching concentration, 1 µg/mL polybrene, 5 mM EDTA and 25–100 µM BB-94. Cells were exposed to HIV-1$_{BAL}$ (~4×10$^5$ TCID$_{50}$) at 1:5000 dilution for 1 h at 37 °C followed by washing with 10 mL culture media. Culture supernatants were sampled and wells were replenished with fresh media on days 6 or 7, or otherwise indicated days p.i. Infections were detected by either intracellular p24 at indicated time points or real-time PCR for viral DNA copy number at day 3 post infections. Intracellular p24 levels were measured on viable CD3+ populations using FITC-conjugated KC57 antibody using the Cytofix/Cytoperm kit from BD Biosciences (San Jose, CA). Samples were collected on an FACSCanto II (BD Biosciences). Real-time PCR was assayed as described previously[67]. All statistical analyses were carried out using the software Prism 7 (GraphPad Software, Inc.). The inhibitors were added 30 min before infection and were replenished after the postinfection wash and subsequent media exchanges. The concentrations of soluble L-selectin in supernatants of infected and uninfected samples were quantified using the human L-selectin DuoSet ELISA kit (R&D Systems) and normalized to per day accumulation. To detect the protease cleaved 6 kD C-terminal membrane-bound fragment of CD62L, western blot analyses were performed similar to previously reported[51]. In brief, equal amount of cell lysates from 10$^7$ HIV-infected or uninfected PBMC in the presence of BB-94, DMDP, or control DMSO were incubated with 1 µg Dynabeads-conjugated anti-CD62L cytoplasmic domain antibody (RayBiotech, cat. no. 119-17273) for 20 min. After three washes, the protein is eluted for SDS-PAGE, and blotted onto a PVDF membrane. The membrane is blocked with 5% nonfat milk for 2 h and then incubated with the anti-CD62L cytoplasmic domain antibody for overnight. After washes, the membrane is probed with HRP-conjugated goat anti-rabbit antibody for 1 h, and visualized with chemiluminescent reagents. As positive controls for the cleaved 6 kD CD62L fragment, uninfected PBMC were activated with Phaseolus vulgaris agglutinin (PHA) for 5 days and treated with phorbol 12-myristate 13-acetate (PMA) for 30 min to induce L-selectin shedding. For trypsin-treated viral release, PBMC were infected with HIV-1$_{BAL}$ in the presence and absence of BB-94. On day 6 of the viral infection, supernatants were collected, and the infected cells were washed with PBS and treated with 1× cell culture trypsin-EDTA solution or media for 15 min. The amount of virus released was determined by p24 ELISA.

**PNGase treatment of activated CD8-depleted PBMC**. HIV-1$_{BAL}$ virus was diluted to 1:5000 in RPMI 1640 containing 0.5% FBS. The virus was treated with 20,000 U PNGase/mL or mock for 1 h at 37 °C. Activated CD8-depleted PBMC were resuspended at 2×10$^6$/mL and infected with either the PNGase-treated, or

mock-treated virus for 1 h at 37 °C as described above. The CD8− PBMC were then washed with RPMI 1640 containing 10% FBS and resuspended at $2 \times 10^6$ cells/mL and plated in a 48-well plate. Infection and analysis as described above.

**Central memory CD4 T-cell staining**. On day 6 or 11 p.i., cells were harvested and stained with T-cell memory surface markers including CD3, CD4, CD27, CD45RO, CD62L, CCR7, or the appropriate isotype controls. Cells were washed, permeabilized using the BD Cytofix/Cytoperm kit (BD Biosciences) according to the manufacturer's instructions and stained for intracellular p24. Samples were acquired on an FACSCanto II and analyzed using FlowJo software (FlowJo, LLC; Ashland, OR). The covariance between $T_{CM}$ and $T_{TM}$ was calculated using two-way ANOVA for p24+ and p24− populations.

**Stimulation for IFNγ, TNF-α, and MIP-1β production**. On day 6 p.i., CD8-depleted PBMC were stimulated for 6 h with Leukocyte Activation Cocktail (BD Biosciences) at 37 °C. Cells were stained for memory cell markers as above, permeabilized with the BD Cytofix/Cytoperm Kit, then washed and stained for intracellular p24, IFNγ, TNF-α, and MIP-1β. Samples were acquired on an FACSCanto II and analyzed using FlowJo software.

**Cell−cell transfer infections**. For the cell−cell transfer-mediated infection, TZM-BL cells were seeded in a 96-well, flat-bottom plate at 3000 cells/well 3 days before the assay. PBMC infected with $HIV-1_{BAL}$ for 3 days with and without the presence of BB-94 were added to the wells at the concentration of $80 \times 10^3$, $40 \times 10^3$, $20 \times 10^3$, $10 \times 10^3$, and $5 \times 10^3$ cells/well in equal volume. Fresh BB-94 was added to cells that received BB-94 treatment for the cell−cell transfer assay. All conditions were prepared in triplicate. The cell mixtures were incubated at 37 °C for 3 days, followed by lysis and measurement of the subsequent luciferase expression as per the manufacturer's instructions.

**HIV-1 release assay**. PBMC were obtained by leukapheresis and ficoll-hypaque centrifugation. CD4+ T cells were isolated using a cell separation system (StemCell Technologies). Cells were cultured with medium alone or with plate-bound anti-CD3 and soluble anti-CD28 antibody in the absence (DMSO) or presence of BB-94 in duplicate for 48 h. The copy number of virion-associated HIV RNA in the above cell culture supernatants was determined using the Cobas Ampliprep/Cobas Taqman HIV-1 Test, Version 2.0 (Roche Diagnostics). The limit of detection for this system is 20 copies/mL.

**Confocal microscopy**. CD4+ T cells were prepared from isolated PBMC using the StemCell EasySep™ Human CD4+T Cell Enrichment Kit. Isolated CD4+ cells were then spun onto Superfrost glass slides using a CytoSpin 3 and CytoSep funnels (Thermo Fisher Scientific, Waltham, MA) at 1000 rpm for 3 min followed by fixation in 90% methanol and stained in 1× PBS with 10% FBS and 0.03% NaN₃. Alexa-647-labeled CD62L antibody and FITC-, PE- or biotin-labeled CD4 antibody were used in a 1:250 dilution for 15 min followed by two washes. For biotin-labeled slides streptavidin-conjugated Alexa-405 was added to the staining mix. Labeled slides were mounted with ProLong® Gold Antifade Reagent (Life Technologies, Grand Island, NY) and sealed after 24 h of curing with nail polish. Images were captured on a Zeiss LSM 780 AxioObserver confocal microscope. A 405 nm diode laser and 633 nm diode laser were used to excite Alexa-405-conjugated CD4 antibody and Alexa-647-conjugated CD62L antibody, respectively.

**gp120-Qdot binding assay**. gp120-QDots were prepared by mixing 12-fold molar excess of monomeric gp120 to Qdot® 625 ITK™ carboxyl quantum dots (Thermo Fisher Scientific) with EDC and NHS in 1× PBS with 10 mM HEPES. The gp120 to Qdot ratio is chosen to be similar to the number of envelop trimers observed on a virion[68]. After 2 h at room temperature and overnight at 4 °C, the reaction is quenched with 1 M Tris (final concentration 300 mM) and stored at 4 °C until used. The final concentration used in the assay was 27pM. Full conjugation of gp120 to the Qdots was examined by SDS-PAGE using the Pierce Silver Stain Kit from Thermo Fisher Scientific (Waltham, MA). Anti-CD4 (RPA-T4), anti-CD62L (DREG-56), and isotype (IgG1) were used at 10, 30, and 10 μg/mL, respectively, as in all other experiments. The prepared gp120-Qdots bound specifically to plate-immobilized soluble L-selectin and CD4 (Fig S1D).

Mock or CD62L-transfected HeLa cells were transferred to ethanol-cleaned eight-well glass coverslip chambers and allowed to adhere for 16−48 h before used. Qdots were added to the HeLa cells and allowed to equilibrate before imaging. Images were collected on an Olympus IX-81 microscope adapted for Total Internal Reflection Fluorescence (TIRF) imaging. A 405 nm diode laser was used to excite Qdots and emission was filtered with a 605/40 band-pass filter before imaging on a Cascade IIB 1024EM CCD camera. Image stacks were deconvoluted with a measured PSF in Huygens Essential software by Scientific Volume Imaging (Hilversum, Netherlands) followed by Qdot recognition and quantification using FIJI imaging software[69] and its 3D object counter. For immobilized protein binding, 100 ng of soluble CD4 or CD62L were adsorbed overnight at room temperature to eight-well sterile glass coverslip chambers. The wells were washed twice before the addition of QDots in 1× PBS containing 10% FBS. After room temperature equilibration for at least 10 min, the QDots were imaged as above.

**Transmission and scanning electron microscopy**. Specimens for TEM were fixed with 2.5% glutaraldehyde in 0.1 M Sorenson's buffer. Samples were post-fixed 1 h with 0.5% osmium tetroxide/0.8% potassium ferricyanide, 1 h with 1% tannic acid and overnight with 1% uranyl acetate at 4 °C. Samples were dehydrated with a graded ethanol series, and embedded in Spurr's resin. Thin sections were cut with a Leica UCT ultramicrotome (Vienna, Austria) stained with 1% uranyl acetate and Reynold's lead citrate prior to viewing at 80 kV on a Hitachi 7500 transmission electron microscope (Hitachi-High Technologies, Tokyo, Japan). Digital images were acquired with an AMT digital camera system (AMT, Chazy, NY). For immunogold labeling, ~$10^6$ cells were washed with 2 ml PBS, centrifuged at $250 \times g$ for 5 min, resuspended in 100 μL of labeling buffer (PBS with 1% BSA), and incubated with 10 μg of mouse antihuman CD62L (clone FMC46, Thermo Fisher) for 1 h on ice. After removal of the primary antibody through a cycle of wash, cells were incubated with 10 μg of secondary goat anti-mouse IgG H&L conjugated to 10 nm gold (Electron Microscopy Sciences) for 1 h on ice, then washed and fixed in 2.5% glutaraldehyde with 0.1 M sodium cacodylate at pH 7.4. For SEM, cells were adhered to silicon chips and fixed with 2.5% glutaraldehyde in 0.1 M Sorenson's buffer overnight at 4 °C. Specimens were post-fixed for 1 h with 1% osmium tetroxide and dehydrated in a graded ethanol series. The samples were critical-point dried under $CO_2$ in a Bal-Tec model cpd 030 dryer (Balzers, Liechtenstein), mounted on aluminum studs, and sputter-coated with 75 angstroms of iridium in a model IBS/TM200S ion beam sputter coater (South Bay Technologies, San Clemente, California). Specimens were viewed at 5 kV in a Hitachi SU-8000 field emission SEM (Hitachi-High Technologies, Tokyo, Japan) using secondary imaging mode. Virion-like particles were identified in SEM images as 50–150 nm size cell surface-associated spherical particles, excluding nodules appearing at the tip of filopodia. Virion-like particles in TEM were further characterized by the presence of capsid. The uninfected samples showed less than ten virion-like particles per image using this criteria.

**Data availability**. All data generated or analyzed during this study are available upon request.

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

## Acknowledgements

We thank Dr. Gary Gilliland of Janssen for kindly providing the OKT3 antibody. We thank Mr. Nathan Max for his help in producing recombinant gp120. This work was

supported by the Intramural AIDS Targeted Antiviral Program (IATAP) fund and Intramural Research of National Institute of Allergy and Infectious Diseases, National Institutes of Health.

## Author contributions

J.K., J.I., Z.Z., J.S., E.F., T.-W. C., and P.S. designed the experiments. J.K., J.I., Z.Z., J.S., G.H., A.C., R.W., M.S., B.H., N.S., D.K., E.F. and T.-W. C. carried out the experiments and analyzed the results. J.K., A.C., T.-W.C., and P.S. wrote the paper. J.A. and S.M. provided reagents and technical guidance.

## Additional information

**Competing interests:** The authors declare no competing interests.

