## [Peer Review File · Nature Communications]

Reviewers' comments:

Reviewer #3 (Remarks to the Author):

This manuscript provides evidence that L-selectin contributes to HIV infection of CD4+ cells. Furthermore, shedding of L-selectin promotes viral release. Overall, the experimental data support the authors' conclusions. The findings suggest new approaches to treating HIV by inhibiting shedding of L-selectin.

Specific comments:

The manuscript is challenging to read and could be improved by editing. The figure legends are sometimes obscure, and the labeling of some figures is confusing. For example, in Figure 1F, the labeling does not distinguish fluid-phase inhibitors from molecules immobilized on the plate surface. The authors do not indicate the number of repeat experiments for each figure and do not define the error bars. Some legends indicate conclusions rather than results.

Figure 1 and Figure S1. The authors need to provide blank sensor controls or other approaches to validate the specificity of binding in the SPR experiments. The text is confusing as is the nature of the recombinant proteins used. Apparently, L-selectin-Fc was used, but the main text does not make this clear. Were the recombinant gp120 proteins used as analytes shown to be monomeric? This is important for interpreting the apparent affinities recorded.

The title is too long and could be condensed to strengthen its impact.

Introduction, bottom of first paragraph, second to last sentence is incorrect. L-selectin does not bind to sulfated sialyl Lewis x on PSGL-1. It binds to sulfated tyrosines plus sialyl Lewis x on an adjacent O-glycan. In contrast, L-selectin binds to sulfated sialylated Lewis x on both N- and O-glycans on HEV mucins in lymph nodes.

Reviewer #4 (Remarks to the Author):

Kononchik et al. present new interesting data that implicate the dual role of CD62L or L-selectin in promoting: 1) HIV adhesion to target cells by interacting with HIV Env and 2) HIV virion release from infected cells that is accompanied by L-selectin shedding.

In this revised manuscript has incorporated additional data that address many of the previous critiques. The findings are clearly significant for HIV research and also for the field of virology in general. However, the author's claim that HIV infection resulted in L-selectin shedding from infected cells is still not supported by experimental data. Moreover, as raised by previous reviewers, this revised manuscript remains poorly written, contains many inaccurate, confusing, or awkward sentences, and thus requires extensive editing. Specific points to be addressed to improve this manuscript are listed below.

The subsection "HIV-1 infection resulted in L-selectin shedding from infected cells" does not reflect the data accurately. The data in Fig 4 A-E demonstrate reduction of CD62L expression but does not show L-selectin shedding from infected cells. Fig 4E does not show correlation between reduction of CD62L+ T cells with increase in the number of infected T cells.

The subsection "L-selectin shedding contributes to the apparent loss of CD4+ central memory T cells" also has the same problem. No data about CD62L shedding are presented in Fig 5.

L-selection shedding is shown directly in Fig S6A and S6C, but no HIV was included in these experiments. Fig 6B includes HIV, but the intensity of 6KD CD62L fragment should be quantified relative to actin band. Actin bands showed variability across the lanes.

The mechanism of L-selectin shedding required for HIV release should be clarified. At least it should be assessed if this process involves L-selectin interaction with Env N-glycans, which the authors evaluated for the virus adhesion/entry part.

Fig 1B: Is the difference between untreated vs PNGase F-treated gp120 significant? Indicate significance by * or ns as done for the other panels. If binding to CD62L is not significantly different, the statement in Results "Treatment of gp120 with peptide....reduced its binding to sCD62L...." should be revised accordingly.

Figures 1-Figure 7: Reproducibility of the results should be indicated. How many times was each experiment repeated? Are the data compiled from all repeat experiments?

For experiments with primary T cells (e.g. Fig 1E, Fig 4), how many donors were tested? How much variability was observed among donors?

Fig 2A: the significant differences should be for comparison of both clones vs CEM. What is CD62L KD clone #3?

Fig 2G and Fig 3: the name of anti-CD62L, DREG-56, should be included in the legend.

Fig 4D: Comparison should also be made with uninfected. Significant and no significant differences should be indicated.

Fig 5B: Statistical significance or no significance should also be included for Tem.

Fig 5C: Does BAL indicate both p24+ and p24- cells from BAL-infected cultures?

Fig 6A needs legend to indicate what are shown by gray area vs blue line?

Fig 7 needs description/criteria for what are considered virions or virus-like particles in SEM and TEM. Methods for SEM and TEM are not included.

Examples of sentences with confusing/imprecise statements:

1st and last sentences in paragraph 1 (Introduction) and last sentence in paragraph 3 (Introduction) need editing. What does “the contribution of N-linked glycan to L-selectin function” exactly mean?

Page 10: “Instead, HIV infection-associated shedding of central memory marker CD62L.... compared to normal effector memory T cells.”

Point-by-point response to the reviewers' comments:

We thank both reviewers for their thorough and constructive comments to our manuscript. The title has been shortened at the suggestion of reviewer 3. A central point raised by reviewer 4 is whether we have shown conclusively HIV infection resulted in L-selectin shedding. We have presented the shedding of L-selectin from CEM cells and PBMC as measured by ELISA in the absence of HIV infections. The reviewer commented the need to show the presence of soluble L-selectin during viral infection. While we believe the fundamental L-selectin shedding mechanism remain the same whether it is induced by activation or infection, the practical reason for not performing ELISA using infected samples was to minimize the infectious waste in P3 biocontainment facility. We have now measured the soluble L-selectin from shedding in HIV infected supernatants using ELISA as suggested by the reviewer. The results showed an enhanced L-selectin shedding in the infected samples and this viral induced shedding was inhibited by the metalloproteinase inhibitor (Fig 4F, 6F). See Response 5 below for further details.

Similarly, we also repeated BB-94 inhibition to L-selectin shedding during HIV infection (Fig 6) by further measuring the soluble L-selectin concentration in infected supernatants in the presence and absence of BB-94. We additionally repeated the shedding western blot showing the 6 kD cleaved L-selectin band in the same infection experiments to address actin band intensity variation, and calculated the statistical significance of normalized band intensities using all data without rejection. See Response 6 below.

We also repeated the BIAcore binding of PNGase F treated gp120 to L-selectin. The previous binding data while showing the trend of decreasing in binding upon deglycosylation, did not reach statistical significance. It is likely due to low concentrations of gp120 used in the binding experiments. We increased the amount of gp120 in the PNGase F treatment and repeated the binding experiments. The binding response of non-treated gp120 increased from ~20-50 RU in the previous manuscript to ~150 RU in the revised manuscript. In addition, we included gp120 from an X4 strain virus, gp120_{SF33} in the PNGase F treatment and BIAcore binding experiment. The changes resulted in significant differences in glycosylated versus deglycosylated gp120 binding to L-selectin. See below Response 8 for further details.

We further assessed the mechanism of L-selectin mediated virion retention during viral release using immuno-EM with a gold labeled CD62L antibody to visualize potential colocalization of L-selectin with budding viruses. The results showed that in the absence of BB-94, virion budding site are generally devoid of gold particles even though labeled gold particles are visible nearby. In the presence of BB-94, however, significant number of gold particles were found colocalized with budding virions at the budding focal point, suggesting potential L-selectin tethering of the budding virion (see below Response 7).

Both reviewers commented about the lack of clarity in various sections of our manuscript and suggested us to improve manuscript writing. We have extensively revised the manuscript to address the confusion statements and improve the overall readability.

The following paragraphs provide a point-by-point response to both reviewers' comments with their comments highlighted in *italic*. Due to added new experiments and the effort to improve the manuscript clarity, some figure numbers have changed in the revised manuscript.

Reviewers' comments:

Reviewer #3 (Remarks to the Author):

Comment 1: *This manuscript provides evidence that L-selectin contributes to HIV infection of CD4+ cells. Furthermore, shedding of L-selectin promotes viral release. Overall, the experimental data support the authors' conclusions. The findings suggest new approaches to treating HIV by inhibiting shedding of L-selectin.*

Specific comments:

The manuscript is challenging to read and could be improved by editing. The figure legends are sometimes obscure, and the labeling of some figures is confusing. For example, in Figure 1F, the labeling does not distinguish fluid-phase inhibitors from molecules immobilized on the plate surface. The authors do not indicate the number of repeat experiments for each figure and do not define the error bars. Some legends indicate conclusions rather than results.

Response 1: We have extensively revised the manuscript to improve the readability, to clarify figure legends and labeling. We revised the materials and methods to state all results were from at least two independent experiments. The error bars and p-values are defined in the Figure 1 legend. Figure legends were revised according to the comments to state their respective results rather than conclusions.

Comment 2: *Figure 1 and Figure S1. The authors need to provide blank sensor controls or other approaches to validate the specificity of binding in the SPR experiments. The text is confusing as is the nature of the recombinant proteins used. Apparently, L-selectin-Fc was used, but the main text does not make this clear. Were the recombinant gp120 proteins used as analytes shown to be monomeric? This is important for interpreting the apparent affinities recorded.*

Response 2: We apologize for the lack of clear statement in the BIAcore method section giving the impression that our binding sensorgrams were not control subtracted. The sensorgrams in Figure 1 and S1 represent blank (control) subtracted from the gp120 binding sensorgrams. Examples of the sensorgrams for HIV BaL gp120 binding to CD62L-Fc (flow cell 2) and to IgG1 control (flow cell 1) prior to control subtraction are included below. IgG1 is immobilized in flow cell 1 as a control for Fc fusion of CD62L. The gp120 binding sensorgrams to the control are of square shape, typical of non-specific binding with fast on and off rates. In contrast, gp120 binding to CD62L-Fc showed slower off rate and did not immediately return to the baseline at the beginning of the off-rate phase. We have clarified in the text that L-selectin-Fc was used in BIAcore binding. Both recombinant gp120_{BAL} and gp120_{SF33} from are monomeric from size exclusion chromatography analysis. We have added both size exclusion chromatograms in revised Figure S1A.

Comment 3: *The title is too long and could be condensed to strengthen its impact.*

Response 3: The title is shortened from: “L-selectin functions as an HIV-1 adhesion receptor on CD4⁺ T cells and the virus induces its shedding for release” to: “HIV-1 targets L-selectin for adhesion and induces its shedding for viral release”

Comment 4: *Introduction, bottom of first paragraph, second to last sentence is incorrect. L-selectin does not bind to sulfated sialyl Lewis x on PSGL-1. It binds to sulfated tyrosines plus sialyl Lewis x on an adjacent O-glycan. In contrast, L-selectin binds to sulfated sialylated Lewis x on both N- and O-glycans on HEV mucins in lymph nodes.*

Response 4: The manuscript is revised accordingly.

Reviewer #4 (Remarks to the Author):

Comment 5: *Kononchik et al. present new interesting data that implicate the dual role of CD62L or L-selectin in promoting: 1) HIV adhesion to target cells by interacting with HIV Env and 2) HIV virion release from infected cells that is accompanied by L-selectin shedding.*

In this revised manuscript has incorporated additional data that address many of the previous critiques. The findings are clearly significant for HIV research and also for the field of virology in general. However, the author’s claim that HIV infection resulted in L-selectin shedding from infected cells is still not supported by experimental data. Moreover, as raised by previous reviewers, this revised manuscript

remains poorly written, contains many inaccurate, confusing, or awkward sentences, and thus requires extensive editing.

Specific points to be addressed to improve this manuscript are listed below.

The subsection “HIV-1 infection resulted in L-selectin shedding from infected cells” does not reflect the data accurately. The data in Fig 4 A-E demonstrate reduction of CD62L expression but does not show L-selectin shedding from infected cells. Fig 4E does not show correlation between reduction of CD62L+ T cells with increase in the number of infected T cells.

Response 5: The reviewer commented that we need to show L-selectin shedding from infected samples in order to conclude HIV infection resulted in the selectin shedding. We have performed the suggested experiment and measured soluble L-selectin present in day 3, 6, and 7 infected supernatants by ELISA. Similar to the data presented in Fig 4, HIV_{BAL} infection downregulated both CD4 and CD62L expressions in infected (p24+) population compared to the controls (p24- and uninfected populations) (new Fig S5A-C). Further, the amount of soluble L-selectin present in infected samples increased progressively on day 3, 6, and 7 post infections and they are significantly higher than those present in the uninfected samples (new Fig 4E).

We also included a new Fig 4F to show a concurrent reduction of CD62L+ T cells and increase in the infected p24+ cells. The original Fig 4E showing the decrease of CD62L+ cells on day 3,6 and 11 p.i. is now in supplemental Fig. S5D.

As mentioned above, we revised extensively the manuscript write-up to improve the readability.

Comment 6: *The subsection “L-selectin shedding contributes to the apparent loss of CD4+ central memory T cells” also has the same problem. No data about CD62L shedding are presented in Fig 5. L-selection shedding is shown directly in Fig S6A and S6C, but no HIV was included in these experiments. Fig 6B includes HIV, but the intensity of 6KD CD62L fragment should be quantified relative to actin band. Actin bands showed variability across the lanes.*

Response 6: As mentioned above, we repeated Fig 4 experiments showing the shedding of L-selectin during HIV infection (new Fig 4E). As to the detection of soluble CD62L shedding in infected CD4+ central memory T cells, it is not technically feasible as various memory populations of infected T cells were separated by using specific fluorescence memory cell markers in flow cytometry. Such experimental design does not permit the incorporation of ELISA.

To address if BB-94 inhibited L-selectin shedding in infected samples (Fig. 6), we repeated the experiment to include soluble L-selectin measurement during HIV infections as suggested. The results showed that BB-94 inhibited the accumulation of soluble L-selectin in infected supernatants (new Fig. 6F). As a result, we replaced previous Fig. 6A, 6B and 6C with the new Fig. 6C, 6E, 6F, and 6G to include the soluble CD62L measurement (the revised Fig 6 is shown below). All four panels correspond to the same infection experiment in duplicates. The original panels are moved to the Supplemental Fig S6.

While the actin bands showed slight decrease in intensities from left to right (original Fig. 6B). However, the change is in the direction further enhance the conclusion. That is no cleaved CD62L was visible in BB-94 treated sample despite its the actin band is stronger than those in DMDP and uninfected samples. Nevertheless, to address if the western blot results are statistical significant, we repeated the western blot analysis using the same infection samples as shown in revised Fig 6C, 6E, and 6F and quantified the 6kD shedding fragment from all western blot analyses against actin bands as suggested. The results are included as Fig 6G and Fig. S6B (below).

Figure 4F. The loss of CD62L⁺ CD4 T cells during HIV-1_{BAL} infection of PBMC (upper panel) is accompanied by the increase in the %p24+ cells (lower panel) at day 6 and 11 p.i.

Figure 6 L-selectin shedding is required for HIV release. A) The inhibition of BB-94 to the shedding of soluble CD62L from parental CEM as well as CD62L-transfected clone #2 and knockdown #8. The CD62L expression of various CEM cells are shown in Figure S4. B) The inhibition of BB-94 to X4-tropic HIV-1_{LAI} infection of various CEM cells. C,D) The inhibition of BB-94 to R5-tropic HIV-1_{BAL} (C) or X4-tropic HIV-1_{LAI} (D) infections of PBMC on day 6 p.i. E) The expression of CD62L during HIV-1_{BAL} infection (Panel C) from uninfected (gray area) and HIV-1_{BAL} infected PBMC in the absence (blue lines) or presence of 100uM BB-94 (red). F) Shedding of L-selectin in the infected and uninfected supernatants during HIV-1_{BAL} infection of PBMC (Panel C) in the presence and absence of BB-94. G) The western blot analysis of a cleaved 6kD C-terminal L-selectin fragment in the presence and absence of BB-94 during HIV-1_{BAL} infection (Panel C) (G). Uninfected PBMC were treated with PMA for 30 minutes to induce L-selectin shedding. Lanes 1 and 2 are cell lysates from the infected samples in the presence of DMSO or BB-94. Lanes 3 and 4 are cell lysates from uninfected samples in the absence or presence of PMA. β-actin (lower panel) is used as loading control. H) The effect of ADAM17 specific inhibitors TAPI-1, TAPI-2 to HIV-1_{BAL} infections of PBMC. I) The infection of PBMCs by JRFL- and SF33-pseudotyped virus in the presence of BB-94 or DMSO. Luciferase activity was measured at day 3 p.i. J) Trypsin-mediated viral release assay. The release of viral particles upon trypsin digestion from day 6 of infected PBMCs in the presence and absence of BB-94 or DMSO was measured by p24 ELISA. K) BB-94 inhibition of Cell-cell transfer mediated HIV-1_{BAL} infections. TZM-BL cells were infected through co-incubation with titrating amount of infected PBMCs in the presence of BB-94 or DMSO. The infection of TZM-BL cells was measured by luciferase activity 48-60 hours p.i.

Comment 7: *The mechanism of L-selectin shedding required for HIV release should be clarified. At least it should be assessed if this process involves L-selectin interaction with Env N-glycans, which the authors evaluated for the virus adhesion/entry part.*

Response 7: The mechanism of L-selectin shedding requirement for HIV release is our current research focus. We speculated in the last section of the manuscript: “While the mechanism regulating the selectin shedding and HIV-1 release remains to be elucidated, our results indicate that efficient release of progeny virus depends on L-selectin shedding on infected T cells. It is likely that budding HIV-1 virions are retained on infected cell surface through gp120 binding in the presence of BB-94, but are released when L-selectin is shed.” It is likely that the cis-interaction between L-selectin and gp120 on the budding virus directly influence HIV release.

The reviewer asked if we could assess the involvement of L-selectin-glycan interaction in HIV release similar to what we did for adhesion and entry. For the viral entry assay, we utilized a pair of HEK 293 originated cell lines, 293T and 293S GnTI⁻ that differ in their glycan processing. We produced JRFL and SF33 pseudoviruses in the two cell lines to generate both glycan deficient JRFL and SF33 viruses. For the viral entry assay, gp120 binds to L-selectin in trans-interaction. For the release assay, however, gp120 is likely to engage L-selectin in cis-interaction (on the same cell as the budding virus), requiring L-selectin expression on the budding cells. Unfortunately, 293T and 293S GnTI⁻ do not express L-selectin, thus our entry approaches are not amendable to viral release. We are experimenting with transfecting L-selectin to these cells and then followed by transfecting with a molecular clone of HIV. These are non-trivial experiments and we are still working on optimizing the parameters (e.g. the dose and timing of the consecutive transfections to optimize L-selectin expression vs viral particle budding) and are not sure if the experimental design is going to work.

However, we further assessed the involvement of L-selectin in HIV budding using immunogold labeled CD62L antibody and examined TEM images of HIV infected cells in the presence of BB-94 for potential L-selectin mediated viral tethering. While gold particles were readily found on infected cells, they were generally not associated with budding virions in the absence of BB-94 (Fig. S7A-B and below). In the presence of BB-94, however, significant number of gold particles were found co-localized with budding virions at the budding focal point (Fig. S7C-D), suggesting the presence of the selectin-mediated

tethering. We replaced the original Fig. S7 showing the SEM and TEM images of uninfected CD4 T cells with the immunogold labeled TEM images to support the potential tethering of budding HIV by L-selectin. The supplemental materials and methods is revised to include immunogold labeling.

Figure S7. Immuno-gold labeled budding HIV viruses. Examples of TEM images of HIV-1_{BAL} budding virions on cells labeled with anti-CD62L primary and gold conjugated secondary antibodies in the absence (A,B) and presence (C,D) of BB-94. Arrows point to the gold particles. Gold particles were generally visible on infected T cells near but not associated with HIV budding virions (A,B) in the absence of BB-94. In the presence of BB-94, however, significant number of budding virions are associated with gold particles, suggesting the tethering of L-selectin to the budding virions.

Comment 8: Fig 1B: Is the difference between untreated vs PNGase F-treated gp120 significant? Indicate significance by * or ns as done for the other panels. If binding to CD62L is not significantly different, the statement in Results “Treatment of gp120 with peptide....reduced its binding to sCD62L....” should be revised accordingly.

Response 8: PNGase F treatment reduced gp120 binding to CD62L. However, the difference in binding did not reach statistical significance in the original manuscript. It is likely due to either incomplete glycan digestion or lower concentration of gp120 used in the BIAcore experiment. We repeated the PNGase F digest of 293T cell expressed gp120_{BAL} and additionally included gp120_{SF33}. The results showed that CD62L bound significantly less to the deglycosylated gp120_{BAL} and gp120_{SF33} than to their glycosylated gp120s. We replaced the original Fig 1B with the new binding data in the revised manuscript.

Comment 9: Figures 1-Figure 7: Reproducibility of the results should be indicated. How many times was each experiment repeated? Are the data compiled from all repeat experiments?

Response 9: Statements of reproducibility are added to the materials and methods and figure legends. It depends on the experiments whether the data were compiled from all repeat experiments or just one typical repeat experiments. In the western blot analysis of the cleaved CD62L band, all repeats are combined without any data rejection since the internal actin band was used to calculate the ratio, as suggested by the reviewer. In the cases involving obvious variation factors, such as donor variation, the data were generally not combined between repeats. All repeat experiments contained either duplicates or triplicates in them and thus are sufficient to derive error and statistical analysis.

Comment 10: For experiments with primary T cells (e.g. Fig 1E, Fig 4), how many donors were tested? How much variability was observed among donors?

Response 10: At least two donors were tested for each experiment described in this manuscript. In all, we estimated cells from >20 donors were used. We do not know the exact donor numbers as our experimental protocol indicated only healthy donors and thus no effort was made in tracking donors. There were clear donor variations in the infection levels. Most donor cells exhibit 20-60% infection rates under experimental infection conditions on day 6-7 post infections by intracellular p24 staining. Rarely, cells from a donor were refractory to experimental infections. Some of the refractory donor cells were likely due to failure in CD3 activation or failure to sufficiently remove CD8 T cells prior to viral infection.

It is also possible some donors carry genetically HIV resistant chemokine receptor CCR5 mutations. If an infection failed, we generally repeat the experiment without investigating further of the reason. In most of cases, even though the infections and inhibitions varied, their trends were generally consistent. An example of donor variation is presented in Fig 7I, in which HIV release data were presented from several infected individuals in the presence of BB-94. The inhibition of BB-94 to the viral release varied from 30-90%, as stated in the last part of the results.

Comment 11: *Fig 2A: the significant differences should be for comparison of both clones vs CEM. What is CD62L KD clone #3?*

Response 11: The p-values for the differences of both clones vs CEM are indicated in the revised Fig. 2A. CD62L KD clone #3 is a CD62L CRISPR/Cas 9 clone that express parental level of CD62L (Fig S4). That is the CRISPR/Cas 9 did not work with respect to the knocking down of CD62L gene expression. However, it serves as an additional control to allow us to compare the differences between clone 8, 32 and 3. The results are similar whether parental CEM or clone 3 is used (Fig 2A). Better description of each clone is provided in the revised manuscript.

Comment 12: *Fig 2G and Fig 3: the name of anti-CD62L, DREG-56, should be included in the legend.*

Response 12: The figure legend is revised accordingly. We removed Fig 2G to improve the manuscript clarity as the results of this figure showing anti-CD62L blocks HIV infection of CEM cell line is somewhat redundant to Fig 3 from PBMC infections.

Comment 13: *Fig 4D: Comparison should also be made with uninfected. Significant and no significant differences should be indicated.*

Response 13: Comparisons with uninfected group were performed and the statistics were added to Figure 4D as suggested. The differences between p24+ and uninfected groups are similar to the differences between p24+ and p24- populations, except for CD62L- population at day 6. The error for uninfected is significantly larger than the other groups as such the increase in p24+ population did not reach statistical significance over the uninfected population.

Comment 14: *Fig 5B: Statistical significance or no significance should also be included for Tem.*

Response 14: Revised as suggested. The differences in different Tem populations are less significant.

Comment 15: *Fig 5C: Does BAL indicate both p24+ and p24- cells from BAL-infected cultures?*

Response 15: Yes.

Comment 16: *Fig 6A needs legend to indicate what are shown by gray area vs blue line?*

Response 16: The gray area and blue lines represent uninfected and infected samples, respectively. We have clarified this in the revised figure legend.

Comment 17: *Fig 7 needs description/criteria for what are considered virions or virus-like particles in SEM and TEM. Methods for SEM and TEM are not included.*

Response 17: Methods for SEM and TEM are included in the revised supplemental information. The scoring criteria for virion-like particles are included in the methods. For both SEM and TEM, the virus-like particles are defined as cell surface associated spherical particles of 50-150nm in size. Nodules at the tip of filopodia were not counted as virion-like particles.

Comment 18: *Examples of sentences with confusing/imprecise statements:*

1st and last sentences in paragraph 1 (Introduction) and last sentence in paragraph 3 (Introduction) need editing. What does “the contribution of N-linked glycan to L-selectin function” exactly mean?

Page 10: “Instead, HIV infection-associated shedding of central memory marker CD62L.... compared to normal effector memory T cells.”

Response 18: Both sentences are confusing, especially the statement on page 10 regarding to the central memory T cells. We have revised the manuscript extensively to improve the readability. As a result, the last sentence in the introduction now reads as: “As these lectin receptors are not expressed on CD4⁺ T cells, consequently, it is not clear if HIV envelope glycans contribute to the viral infection of T cells despite earlier studies showing some gp120 glycan mutations resulted in replication deficient virus.”

The page 10 statement is revised to: “We speculate that the previously reported T_{EM} dysfunction may result from grouping of infected T_{TM}, the CD62L shedded T_{CM}, as T_{EM} cells based on L-selectin expression. Since T_{CM} produce less cytokines than T_{EM} cells, this would result in an apparent decrease in cytokine production in T_{EM} cells.”

REVIEWERS' COMMENTS:

Reviewer #1 (Remarks to the Author):

The revised manuscript is significantly improved, although there are still frequent errors that could be corrected with better editing. Moreover, there are two sets of figure legends that in the manuscript file, which are not identical. Some figure legends still do not indicate the number of replicates, particularly for images where a representative result is shown.

Reviewer #2 (Remarks to the Author):

The authors have made significant revisions to address each of the concerns raised in the previous reviews. New data are now presented to better support the conclusion. English usage is also much improved.

Minor comments:

- the term of "glycan-deficient" to refer to 293S-produced viruses or gp120 proteins is not correct. Complex glycan-deficient should be more accurate.
- The differences seen in Fig 5B with Tem are minimal but they are significant for day 6 (*) and not significant for day 11 (ns). The statement in line 168-170 should be revised.

Point-by-point response:

Reviewer #1 (Remarks to the Author):

Comment 1: *The revised manuscript is significantly improved, although there are still frequent errors that could be corrected with better editing.*

Response: The manuscript has been edited in detail by two English-speaking native authors for grammatical errors. As a result, we corrected more than a dozen such errors in the revised manuscript.

Comment 2: *Moreover, there are two sets of figure legends that in the manuscript file, which are not identical. Some figure legends still do not indicate the number of replicates, particularly for images where a representative result is shown.*

Response: I apologize the slight wording differences in the two sets of figure legends due to inconsistency in transferring the revised figure legends to the second set, the one associated with the figures is to help reviewers to navigate the figures. We only include one set of figure legends in this revised submission for publication.

We included statement of replicates for images shown as representatives in Fig 7A-F, as well as for all Supplementary Figures.

Reviewer #2 (Remarks to the Author):

The authors have made significant revisions to address each of the concerns raised in the previous reviews. New data are now presented to better support the conclusion. English usage is also much improved.

Minor comments:

Comment 1: *- the term of "glycan-deficient" to refer to 293S-produced viruses or gp120 proteins is not correct. Complex glycan-deficient should be more accurate.*

Response: changes have been made as suggested.

Comment 2: *- The differences seen in Fig 5B with Tem are minimal but they are significant for day 6 (*) and not significant for day 11 (ns). The statement in line 168-170 should be revised.*

Response: We revised the mentioned statement from

its original sentence: "In contrast, the number of CD45RO⁺/CD27⁻ effector memory (T_{EM}) cells remains similar in the p24⁺ compared to the p24⁻ population on both days 6 and 11 (Fig. 5B)."

to: "In comparison, a smaller increase in the number of CD45RO⁺/CD27⁻ T_{EM} cells was observed between the p24⁺ and p24⁻ populations (Fig. 5B)."